

# The quasi-equilibrium framework re-visited: analyzing long-term CO₂ enrichment responses in plant-soil models

Mingkai Jiang[1], Sönke Zaehle[2], Martin G. De Kauwe[3], Anthony P. Walker[4], Silvia Caldararu[2], David S. Ellsworth[1], Belinda E. Medlyn[1]

[1]Hawkesbury Institute for the Environment, Western Sydney University, Locked Bag 1797, Penrith, NSW, Australia, 2751

[2]Max Planck Institute of Biogeochemistry, Jena, Germany

[3]ARC Centre of Excellence for Climate Extremes, University of New South Wales, Sydney, NSW 2052,
Australia

[4]Environmental Sciences Division and Climate Change Science Institute, Oak Ridge National Laboratory, Oak Ridge, TN, 37831, USA

*Correspondence to*: Mingkai Jiang (m.jiang@westernsydney.edu.au)

**Abstract.** Elevated carbon dioxide (CO₂) can increase plant growth, but the magnitude of this CO₂ fertilization effect is modified by soil nutrient availability. Predicting how nutrient availability affects plant responses to elevated CO₂ is a key consideration for ecosystem models, and many modelling groups
have moved to, or are moving towards, incorporating nutrient limitation in their models. The choice of assumptions to represent nutrient cycling processes has a major impact on model predictions, but it can be difficult to attribute outcomes to specific assumptions in complex ecosystem simulation models. Here we revisit the quasi-equilibrium (QE) analytical framework introduced by Comins & McMurtrie (1993) and explore the consequences of specific model assumptions for ecosystem net primary productivity. We
review the literature applying this framework to plant-soil models, and then examine the effect of several new assumptions on predicted plant responses to elevated CO₂. Examination of alternative assumptions



for plant nitrogen uptake showed that a linear function of the mineral nitrogen pool or a saturating function of root biomass yield similar $CO_2$ responses over time. In contrast, a saturating function of the mineral nitrogen pool yields no soil nutrient feedback at the very long-term, near-equilibrium timescale, meaning that a full $CO_2$ fertilization effect on production is realized. We show that incorporating a priming effect

on slow soil organic matter decomposition attenuates the nutrient feedback effect on production, leading to a strong medium-term $CO_2$ response. Finally, we demonstrate that using a "potential NPP" approach to represent nutrient limitation of growth yields a relatively small $CO_2$ fertilization effect across all timescales. Our results highlight that the QE analytical framework is effective for evaluating both the consequence and the mechanism through which different model assumptions affect predictions. To help

constrain predictions of the future terrestrial carbon sink, we recommend use of this framework to analyze likely outcomes of new model assumptions before introducing them to complex model structures.

**Keywords**: analytical approximation | equilibrium | $CO_2$ fertilization | nitrogen | priming | nutrient uptake

## 1 Introduction

Predicting how plants respond to atmospheric carbon dioxide ($CO_2$) enrichment ($eCO_2$) under nutrient limitation is fundamental for an accurate estimate of the global terrestrial carbon (C) budget in response to climate change. There is now ample evidence that the response of terrestrial vegetation to $eCO_2$ is modified by soil nutrient availability (Fernández-Martínez et al., 2014; Norby et al., 2010; Reich and Hobbie, 2012; Sigurdsson et al., 2013). Over the past decade, land surface models have developed from

C-only models to carbon-nitrogen (CN) models (Gerber et al., 2010; Zaehle and Friend, 2010). The inclusion of C-N biogeochemistry has been shown to be essential to capture the reduction in the $CO_2$ fertilization effect with declining nutrient availability and therefore its implications for climate change (Zaehle et al., 2015). However, it has also been shown that models incorporating different assumptions predict very different vegetation responses to $eCO_2$ (Lovenduski and Bonan, 2017; Medlyn et al., 2015).

Careful examination of model outputs has provided insight into the reasons for the different model





predictions (De Kauwe et al., 2014; Medlyn et al., 2016; Walker et al., 2014; Walker et al., 2015; Zaehle et al., 2014), but it is generally difficult to attribute outcomes to specific assumptions in these plant-soil models that differ in structural complexity and process feedbacks (Lovenduski and Bonan, 2017; Medlyn et al., 2015; Thomas et al., 2015).

Understanding the mechanisms underlying predictions of ecosystem carbon cycle processes is fundamental for the validity of prediction across space and time. Comins and McMurtrie (1993) developed an analytical framework, the "quasi-equilibrium" (QE) approach, to make model predictions traceable to their underlying mechanisms. The approach is based on the two-timing approximation method (Ludwig et al., 1978) and makes use of the fact that ecosystem models typically represent a series
of pools with different equilibration times. The method involves: 1) choosing a time interval ($\tau$) such that the model variables can be divided into "fast" pools (which approach effective equilibrium at time $\tau$) and "slow" pools (which change only slightly at time $\tau$); 2) holding the "slow" pools constant, and calculating the equilibria of the "fast" pools (an effective equilibrium as this is not a true equilibrium of the entire system); and 3) substituting the "fast" pool effective equilibria into the original differential equations to
give simplified differential equations for the slow pools at time $\tau$.

In a CN model, plant net primary production (NPP) can be estimated from two constraints, based on equilibration of the C balance (the "photosynthetic constraint") and the N balance (the "nitrogen recycling constraint") (Comins and McMurtrie, 1993). Both constraints link NPP with leaf chemistry (i.e. N:C ratio) (derivation in Section 3.1). The simulated production occurs at the intersection of these two constraint
curves (shown graphically in Figure 1). To understand behaviour on medium and long-time scales (e.g. wood, and slow and passive soil organic pools in Figure 2, 20 – 200 years), one can assume that plant pools with shorter equilibration times in the model (e.g. foliage, fine-root or active soil organic pools in Figure 2) have reached quasi-equilibrium, and model dynamics are thus driven by the behaviour of the longer timescale pools.

The recent era of model development has seen some significant advances in representing complex plant-soil interactions, but models still diverge in future projections of $CO_2$ fertilization effect on NPP (Friend



et al., 2014; Koven et al., 2015; Walker et al., 2015). A recent series of multi-model inter-comparison studies has demonstrated the importance of understanding underlying response mechanisms in determining model response to future climate change (Medlyn et al., 2015), but this can be difficult to achieve in complex global models. The QE framework is a relatively simple but quantitative method to

examine the effect of different assumptions on model predictions. As such, it complements more computationally expensive sensitivity analyses, and can be used as an effective tool to provide *a priori* evaluation of both the consequence and mechanism through which different new model implementations affect model predictions.

Here, by constructing a QE framework based on the structure of the Generic Decomposition and Yield

(G'DAY) model (Comins and McMurtrie, 1993), we evaluate the effects on plant responses to eCO$_2$ of some recently-developed model assumptions incorporated into ecosystem models, for example the Community Land Model (CLM)(Oleson et al., 2004), the Community Atmosphere–Biosphere Land Exchange (CABLE) model (Kowalczyk et al., 2006), the Lund-Potsdam-Jena (LPJ) model (Smith et al., 2001), the JSBACH model (Goll et al., 2017b), and the O-CN model (Zaehle et al. 2010). Specifically,

we test how different functions affecting plant N uptake influence NPP responses to eCO$_2$ at various quasi-equilibrium time steps. The present study is a continuation of the series of the QE studies as reviewed in Section 2, with a general aim of helping to understanding the similarities and differences of predictions made by different process-based models, as demonstrated in Section 3.

## 20  2. Literature Review

Many of the assumptions currently being incorporated into CN models have previously been explored using the QE framework; here we provide a brief literature review describing the outcomes of this work (Table 1). Firstly, the flexibility of plant and soil stoichiometry has recently been highlighted as a key assumption (Stocker et al., 2016; Zaehle et al., 2014). A key finding from early papers applying the QE

framework was that model assumptions about the flexibility of the plant wood N:C ratio (Comins, 1994;





Comins and McMurtrie, 1993; Dewar and McMurtrie, 1996; Kirschbaum et al., 1994; Kirschbaum et al., 1998; McMurtrie and Comins, 1996; Medlyn and Dewar, 1996) and soil N:C ratio (McMurtrie and Comins, 1996; McMurtrie et al., 2001; Medlyn et al., 2000) were critical determinants of the magnitude of the transient (10 to > 100 years) plant response to eCO$_2$ (Figure 1). Different to the effect of foliar N:C

ratio flexibility, which has an instantaneous effect on photosynthesis, the flexibility of the wood N:C ratio controls the flexibility of nutrient storage per unit biomass accumulated in the slow turnover pool. Therefore, a constant wood N:C ratio, such as was assumed in CLM4 (Thornton et al., 2007; Yang et al., 2009), means that effectively a fixed amount of N is locked away from the active processes such as photosynthesis on the timescale of the lifespan of the woody tissue. In contrast, a flexible wood N:C ratio,

such as was tested in O-CN (Meyerholt and Zaehle, 2015), allows variable N storage in the woody tissue, and consequently more nutrient available for C uptake at the transient timescale. Similarly, flexibility in the soil N:C ratio determines the degree of the soil N cycle feedback (e.g. N immobilization and mineralization) and therefore its effect on plant response to eCO$_2$. A large response to eCO$_2$ occurs when the soil N:C ratio is allowed to vary, whereas there could be little or no response if the soil N:C ratio is

assumed to be inflexible (McMurtrie and Comins, 1996).

Changes in plant allocation with eCO$_2$ are also a source of disagreement among current models (De Kauwe et al. 2014). The QE framework has been used to investigate a number of different plant C allocation schemes (Comins and McMurtrie, 1993; Kirschbaum et al., 1994; Medlyn and Dewar, 1996). For example, Medlyn and Dewar (1996) suggested that plant long-term growth responses to eCO$_2$ depend

strongly on the extent to which stem and foliage allocations are coupled. With no coupling (i.e. fixed allocation of C and N to stemwood), plant growth was not responsive to eCO$_2$; with linear coupling (i.e. allocation to stemwood proportional to foliage allocation), a significant long-term increase in total growth following eCO$_2$ was found (Figure S1). The reason for this is similar to the argument behind wood N:C ratio flexibility, that decreasing C allocation to wood decreases the rate of N removal per unit of C

invested in growth. In contrast, Kirschbaum et al. (1994) found that changes in allocation between different parts of plant only marginally changed the CO$_2$ sensitivity of production at different timescales.





The fundamental difference between the two allocation schemes was that Kirschbaum et al. (1994) assumed that the root allocation coefficient was determined by a negative relationship with the foliar N:C ratio, meaning that the increase in foliar N:C ratio would lead to a decreased root allocation and increased wood and foliage allocation, whereas Medlyn and Dewar (1996) investigated stem-foliage allocation

coupling without introducing a feedback via the foliar N:C ratio. The comparison of the two allocation schemes is indicative of the underlying causes of model prediction divergence in recent inter-model comparisons (De Kauwe et al., 2014; Walker et al., 2015).

Another hypothesis currently being explored in models is the idea that increased belowground allocation can enhance nutrient availability under elevated $CO_2$ (Dybzinski et al., 2014; Guenet et al., 2016). Comins

(1994) argued that the N deficit induced by $CO_2$ fertilization could be eliminated by stimulation of N fixation. This argument was explored in more detail by McMurtrie et al. (2000), who assumed that $eCO_2$ led to a shift in allocation from wood to root exudation, which resulted in enhanced N fixation. They showed that, although the increase in N fixation could induce a large $eCO_2$ response in NPP over the long-term, a slight decrease in NPP was predicted over the medium-term. This decrease occurred because

increased exudation at $eCO_2$ increased soil C input, causing increased soil N sequestration and lowering the N available for plant uptake. Over the long-term, however, both NPP and C storage were greatly enhanced because the sustained small increase in N input led to a significant build-up in total ecosystem N on this timescale.

The interaction between rising $CO_2$ and warming under nutrient limitation is of key importance for future

simulations. Medlyn et al. (2000) demonstrated that short-term plant responses to warming, such as physiological acclimation, are over-ridden by the positive effects of warming on soil nutrient availability in the medium to long term. Similarly, McMurtrie et al. (2001) investigated how the flexibility of the soil N:C ratio affects predictions of the future C sink under elevated temperature and $CO_2$. They showed that assuming an inflexible soil N:C ratio with elevated temperature would mean a release of nitrogen with

enhanced decomposition, leading to a large plant uptake of N to enhance growth. In contrast, an inflexible soil N:C ratio would mean that the extra N mineralized under elevated temperature is largely immobilized





in the soil and hence a smaller increase in C storage. This effect of soil N:C stoichiometry on the response to warming is opposite to the effect on $eCO_2$ described above. Therefore, under a scenario where both temperature and $CO_2$ increase, the C sink strength is relatively insensitive to soil N:C variability, but the relative contributions of temperature and $CO_2$ to this sink differ under different soil N:C ratio assumptions

(McMurtrie et al., 2001). This outcome may explain the results observed by Bonan and Levis (2010) when comparing coupled carbon cycle-climate simulations. The TEM (Sokolov et al., 2008) and CLM models (Thornton et al., 2009), which assumed inflexible stoichiometry, had a large climate-carbon feedback but a small concentration-carbon feedback, contrasting with the O-CN model (Zaehle et al., 2010), which assumed flexible stoichiometry and had a small climate-carbon feedback and a large

concentration-carbon feedback. Variations among models in this stoichiometric flexibility assumption could potentially also explain the trade-off between $CO_2$ and temperature sensitivities observed by Huntzinger et al. (2017).

### 3. Methods and Results

Below we first describe the baseline simulation model and derivation of the QE constraints (Section 3.1),

then follow with subsections on each of the new model assumptions tested in this study (Sections 3.2 – 3.4). Within each subsection, we first provide key equations for each assumption and the derivation of the QE constraints with these new assumptions, then provide our graphic interpretations and analyses to understand the effect of the model assumption on plant NPP responses to $eCO_2$.

Here we tested alternative model assumptions for three processes that affect plant carbon-nitrogen

cycling: (1) different ways of representing plant N uptake, namely plant N uptake as a fixed fraction of the mineral N pools, as a saturating function of the mineral N pool (Zaehle and Friend, 2010), or as a saturating function of root biomass (McMurtrie et al., 2012); (2) the "potential NPP" approach that downregulates potential NPP to represent N limitation (Oleson et al., 2004); and (3) root exudation and its effect on soil organic matter decomposition rate (i.e. priming effect). The first two assumptions have

been incorporated into some existing land surface model structures (e.g. CLM, CABLE, O-CN, LPJ),



whereas the third is a framework proposed following the observation that models did not simulate some key characteristic observations of the DukeFACE experiment (Walker et al., 2015; Zaehle et al., 2014), and therefore could be of importance in addressing some model limitations in representing soil processes (van Groenigen et al., 2014; Zaehle et al., 2014). Here we do not target specific ecosystems to

parameterize the model but anticipate that the analytical interpretation of QE framework is of general applicability for woody-dominated ecosystems.

### 3.1 Baseline model and derivation of the QE constraints

Our baseline simulation model is similar in structure to G'DAY (Generic Decomposition And Yield, Comins & McMurtrie 1993), a generic ecosystem model that simulates biogeochemical processes (C, N,

and $H_2O$) at daily or sub-daily time steps. A simplified G'DAY model version that simulates plant-soil C-N interactions at a weekly timestep was developed for this study (Figure 2). In G'DAY, plants are represented by three stoichiometrically flexible pools: foliage, wood and roots. Each pool turns over at a fixed rate. Litter enters one of four litter pools (metabolic and structural above- and below-ground) and decomposes at a rate dependent on the litter N:C ratio, soil moisture and temperature. Soil organic matter

(SOM) is represented as active, slow and passive pools, which decay according to first order decay functions with different rate constants. Plants access nutrients from the mineral N pool, which is an explicit pool supplied by SOM decomposition and an external input, which is assumed to be constant, as a simplified representation of fixation and atmospheric deposition.

Gross primary production (GPP) is calculated using a light-use efficiency approach named MATE (Model

Any Terrestrial Ecosystem) (McMurtrie et al., 2008; Medlyn et al., 2011; Sands, 1995), in which absorbed photosynthetically active radiation is estimated from leaf area index ($L$) using Beer's Law, and is then multiplied by a light-use efficiency (LUE) which depends on the foliar N:C ratio ($n_f$) and atmospheric $CO_2$ concentration ($C_a$).

$$GPP = LUE(n_f, C_a) \cdot I_0 \cdot (1 - e^{-kL}) \qquad \text{(Eq. 1)}$$





where $I_0$ is the incident radiation, $k$ is the canopy light extinction coefficient, and $L$ is leaf area index. The derivation of LUE for the MATE model is described in full by McMurtrie et al. (2008); our version differs only in that the key parameters determining the photosynthetic rate follow the empirical relationship with foliar N:C ratio given by Walker et al. (2014a) and the expression for stomatal conductance follows
Medlyn et al. (2011).

The baseline simulation model further assumes that: 1) carbon use efficiency (the ratio of NPP:GPP) is constant; 2) allocation of newly fixed carbon among foliage, wood and root pools is constant; 3) foliage, wood and root N:C ratios are flexible; 4) wood and root N:C ratios are proportional to the foliar N:C ratio, with constants of proportionality $r_w$ and $r_r$, respectively 5) a constant proportion ($t_f$) of foliage N is
retranslocated before leaves senesce; 6) active, slow and passive SOM pools have fixed N:C ratios; and 7) an N uptake constant determines the plant N uptake rate. Definitions of parameters and forcing variables are summarized in Table 2. For all simulations, ambient $CO_2$ concentration (a$CO_2$) was set at 400 ppm and e$CO_2$ at 800 ppm.

We now summarize the derivation of the two QE constraints, the photosynthetic constraint and the
nutrient cycling constraint, from our baseline simulation model. The derivation follows Comins and McMurtrie (1993), which is further elaborated in work by (McMurtrie et al., 2000; Medlyn and Dewar, 1996), and evaluated (Comins, 1994). First, the photosynthetic constraint is derived by assuming that the foliage C pool ($C_f$) has equilibrated. That is, the new foliage C production equals turnover, which is assumed to be a constant fraction ($s_f$) of the pool:

$$a_f NPP = s_f C_f \qquad \text{(Eq. 2)}$$

where $a_f$ is the allocation coefficient for foliage. From Eq. 1, net primary production is a function of the foliar N:C ratio and the foliage C pool:

$$NPP = LUE(n_f, C_a) \cdot I_0 \cdot \left(1 - e^{-k\sigma C_f}\right) \cdot CUE \qquad \text{(Eq. 3)}$$





Where $\sigma$ is the specific leaf area. Combining two equations above leads to an implicit relationship between $NPP$ and $n_f$:

$$NPP = LUE(n_f, C_a) \cdot I_0 \cdot \left(1 - e^{-k\sigma a_f NPP/s_f}\right) \cdot CUE \qquad \text{(Eq. 4)}$$

which is the photosynthetic constraint.

Secondly, the nitrogen cycling constraint is derived by assuming that nitrogen inputs to, and outputs from,
the equilibrated pools, are equal. Based on the assumed residence times of the passive SOM (~400 years), slow SOM (15 years) and woody biomass (50 years) pools, we can calculate the nutrient recycling constraint at three different timescales: very long (VL, > 500 years, all pools equilibrated), long (L, 100 – 500 years, all pools equilibrated except the passive pool), or medium (M, 5-50 years, all pools equilibrated except slow, passive and wood pools). At the VL-term, we have:

$$N_{in} = N_{loss} \qquad \text{(Eq. 5)}$$

where $N_{in}$ is the total N input into the system, and $N_{loss}$ is the total N lost from the system via leaching and volatilisation. Following Comins and McMurtrie (1993), the flux $N_{in}$ is assumed to be a constant. The total N loss term is proportional to the rate of N mineralization ($N_m$), following:

$$N_{loss} = l_n \cdot N_m \qquad \text{(Eq. 6)}$$

where $l_n$ is the fraction of N mineralization that is lost. It is assumed that mineralised N that is not lost is taken up by plants ($N_U$):

$$N_U = N_m - N_{loss} \qquad \text{(Eq. 7)}$$

Combining with Eq. 6, we have:

$$N_{loss} = \frac{l_n}{(1 - l_n)} N_U \qquad \text{(Eq. 8)}$$

The plant N uptake rate depends on production (NPP) and plant N:C ratios, according to:



$$N_U = NPP \cdot (a_f n_{fl} + a_w n_w + a_r n_r) \qquad \text{(Eq. 9)}$$

Where $a_f$, $a_w$ and $a_r$ are the allocation coefficients for foliage, wood and roots, respectively, and $n_{fl}$, $n_w$ and $n_r$ are the N:C ratios for foliage litter, wood and roots, respectively. Foliage litter N:C ratio ($n_{fl}$) is proportional to $n_f$, according to Table 2. Combining Eq. 9 with Eq. 5 and Eq. 8, we obtain a function of NPP that can be related to total N input, which is the nutrient recycling constraint at the VL-term,

expressed as:

$$NPP = \frac{N_{in}(1 - l_n)}{l_n(a_f n_{fl} + a_w n_w + a_r n_r)} \qquad \text{(Eq. 10)}$$

Since $n_w$ and $n_r$ are assumed proportional to $n_f$, the nutrient recycling constraint also links NPP and $n_f$. The intersection with the photosynthetic constraint yields the very-long term equilibria of both NPP and $n_f$.

At the L-term, we now have to consider N flows leaving and entering the passive SOM pool, which is no

longer equilibrated:

$$N_{in} + N_{R_p} = N_{loss} + N_{S_p} \qquad \text{(Eq. 11)}$$

where $N_{R_p}$ and $N_{S_p}$ are the release and sequestration of the passive SOM N pool, respectively. The release flux, $N_{R_p}$, can be assumed to be constant on the L-term timescale. The sequestration flux, $N_{S_p}$, can be calculated as a function of NPP. In G'DAY, as with most carbon-nitrogen coupled ecosystem models, carbon flows out of the soil pools are directly related to the pool size. As demonstrated by Comins and

McMurtrie (1993), such soil models have the mathematical property of linearity, meaning that carbon flows out of the soil pools are proportional to the production input to the soil pool, or NPP. Furthermore, the litter input into the soil pools is assumed proportional to foliar N:C ratio, with the consequence that N sequestered in the passive SOM is also related to foliar N:C ratio. The sequestration flux into the passive soil pool ($N_{S_p}$) can thus be written as:



$$N_{S_p} = NPP \, n_p (\Omega_{p_f} \cdot a_f + \Omega_{p_w} \cdot a_w + \Omega_{p_r} \cdot a_r) \qquad \text{(Eq. 11)}$$

Where $n_p$ is the N:C ratio of the passive SOM pool, $\Omega_{p_f}$, $\Omega_{p_w}$ and $\Omega_{p_r}$ are the burial coefficients for foliage, wood and roots (the proportion of plant carbon production that is ultimately buried in the passive pool), respectively. The burial coefficients $\Omega_{p_f}$, $\Omega_{p_w}$ and $\Omega_{p_r}$ depend on the N:C ratios of foliage, wood and root litter (detailed derivation in Comins and McMurtrie, 1993). Combining and re-arranging, we

5  obtain nutrient recycling constraint at the L-term as:

$$NPP = \frac{N_{in} + N_{R_p}}{n_p \left( \Omega_{p_r} a_r + \Omega_{p_f} a_f + \Omega_{p_w} a_w \right) + \dfrac{l_n}{1 - l_n} (a_f n_{fl} + a_w n_w + a_r n_r)} \qquad \text{(Eq. 13)}$$

Similarly, at the M-term, we have:

$$N_{in} + N_{R_p} + N_{R_s} + N_{R_w} = N_{loss} + N_{S_p} + N_{S_s} + N_{S_w} \qquad \text{(Eq. 14)}$$

Where $N_{R_s}$ and $N_{R_w}$ are the N released from slow SOM and wood pool, respectively, and $N_{S_s}$ and $N_{S_w}$ are the N stored in slow SOM and wood pool, respectively (Medlyn et al., 2000). The nutrient recycling constraint at the M-term can thus be derived as:

$$NPP \qquad \qquad \qquad \qquad \qquad \qquad \qquad \qquad \qquad \qquad \qquad \qquad \text{(Eq. 15)}$$
$$= \frac{N_{in} + N_{R_p} + N_{R_s} + N_{R_w}}{a_f \left( \Omega_{s_f} n_s + \Omega_{p_f} n_p \right) + a_r \left( \Omega_{s_r} n_s + \Omega_{p_r} n_p \right) + \dfrac{l_n}{1 - l_n} (a_f n_{fl} + a_w n_w + a_r n_r) + a_w n_w}$$

10  Where $n_s$ is the slow SOM pool N:C ratio, $\Omega_{s_f}$ and $\Omega_{s_r}$ are foliage and root C sequestration rate into slow SOM pool, respectively (Medlyn et al., 2000).





### 3.2 Explicit plant N uptake

We now move to considering new model assumptions. We first consider different representations of plan N uptake. In the baseline model, the mineral N pool ($N_{min}$) is implicit, as we assumed that all mineralized N in the soil is either taken up by plants ($N_U$) or lost from the system ($N_{loss}$). Here, we evaluate three alternative model representations where plant N uptake depends on an explicit $N_{min}$ pool, and their effects on plant responses to eCO$_2$. We consider plant N uptake as 1) a fixed coefficient of the mineral N pool, 2) a saturating function of root biomass and a linear function of the mineral N pool (McMurtrie et al., 2012), and 3) a saturating function of the mineral N pool and a linear function of root biomass. The last function has been incorporated into some land surface models, for example, O-CN (Zaehle and Friend, 2010) and CLM (Ghimire et al., 2016), while the first two have been incorporated into G'DAY(Corbeels et al., 2005).

A mineral N pool was made explicit by specifying a constant coefficient ($u$) to regulate the plant N uptake rate (i.e. $N_U = u \cdot N_{min}$). N lost from the system is a function of mineral N pool ($N_{min}$), regulated by a loss rate ($l_{n,rate}$, yr$^{-1}$). For the VL term equilibrium, we have $N_{in} = N_{loss}$, which means $N_{min} = \frac{N_{in}}{l_{n,rate}}$, hence:

$$N_{loss} = \frac{l_{n,rate}}{u} \cdot NPP \cdot (a_f n_{fl} + a_w n_w + a_r n_r) \qquad \text{(Eq. 16)}$$

Where $n_{fl}$ is the foliage litter N:C ratio, which is proportional to $n_f$ (Table 2). At the VL equilibrium, we can re-arrange the above equation to relate NPP to $n_f$:

$$NPP = \frac{u\,N_{in}}{l_n \cdot (a_f n_{fl} + a_w n_w + a_r n_r)} \qquad \text{(Eq. 17)}$$

which indicates that the N-cycling constraint for NPP is inversely dependent on $n_f$.

The second function represents plant N uptake as a saturating function of root biomass ($C_r$), and a linear function of the mineral N pool (McMurtrie et al., 2012), expressed as:



$$N_U = \frac{C_r}{C_r + K_r} \cdot N_{min} \qquad \text{(Eq. 18)}$$

where $K_r$ is a constant. At the VL equilibrium, we have $N_{in} = N_{loss} = l_{n,rate} N_{min}$, and $C_r = \frac{NPP \cdot a_r}{s_r}$, where $s_r$ is the lifetime of root. Substituting for $C_r$ in Eq. 18, we relate $N_u$ with NPP:

$$N_U = \frac{NPP \cdot a_r}{NPP \cdot a_r + K_r \cdot s_r} \cdot \frac{N_{in}}{l_{n,rate}} \qquad \text{(Eq. 19)}$$

Since $N_U$ is also a function of NPP, we can re-arrange and get:

$$NPP = \frac{N_{in}}{l_{n,rate}\left(a_f n_{fl} + a_w n_w + a_r n_r\right)} - \frac{K_r s_r}{a_r} \qquad \text{(Eq. 20)}$$

Comparing with Eq. 17, here NPP is also inversely dependent on $n_{\text{f}}$, but with an additional negative offset

5   of $\frac{K_r s_r}{a_r}$. The third approach to represent N uptake (e.g. O-CN and CLM) expresses N uptake as a saturating function of mineral N, and also linearly depends on root biomass (Zaehle and Friend, 2010), according to:

$$N_U = \frac{N_{min}}{N_{min} + K} \cdot C_r \cdot V_{max} \qquad \text{(Eq. 21)}$$

where $K$ is a constant coefficient, and $V_{max}$, the maximum root N uptake capacity, is simplified as a constant here. Since $N_U$ is also a function of NPP, we get

$$N_{min} = K \cdot \frac{\left(a_f n_{fl} + a_w n_w + a_r n_r\right)}{V_{max} \cdot \frac{a_r}{s_r} - \left(a_f n_{fl} + a_w n_w + a_r n_r\right)} \qquad \text{(Eq. 22)}$$


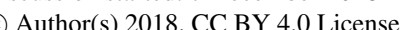


This equation sets a limit to possible values of $n_f$. In equilibrium, for $N_{min}$ to be non-zero, we need $\left(a_f n_{fl} + a_w n_w + a_r n_r\right) < V_{max} \frac{a_r}{s_r}$. The N loss rate is still proportional to the mineral N pool, so $N_{loss}$ is given by

$$N_{loss} = l_{n,rate} \cdot K \cdot \frac{\left(a_f n_{fl} + a_w n_{wl} + a_r n_{rl}\right)}{V_{max} \cdot \frac{a_r}{s_r} - \left(a_f n_{fl} + a_w n_{wl} + a_r n_{rl}\right)} \qquad \text{(Eq. 23)}$$

The above equation provides a $N_{loss}$ term that no longer depends on NPP, but only on $n_f$. If the N

leaching loss is the only system N loss, the VL-term nutrient constraint no longer involves NPP, implying that the full photosynthetic $CO_2$ fertilization effect is realized. The L- and M-term nutrient recycling constraints, however, are still NPP-dependent, due to feedbacks from the slowly recycling wood and SOM pools (e.g. Eq. 11 – 15).

The impacts of these alternative representations of N uptake are shown in Figure 4. First, the explicit

consideration of the mineral N pool with a fixed uptake constant ($u$) of 1 yr$^{-1}$ has little impact on the transient response to eCO$_2$ when compared to the baseline model (Figure 4a, Figure 1a, Table 3). Varying $u$ does not strongly (<5%) affect plant responses to CO$_2$ fertilization at different time steps (Figure S2). This is because $u$ is only a scaling factor of NPP, meaning it affects NPP but not its response to eCO$_2$ (Table 4), as depicted by Eq. 17.

Moreover, the approach that assumes N uptake as a saturating function of root biomass (McMurtrie et al., 2012) has comparable eCO$_2$ effects on production to the baseline and the fixed uptake coefficient models (Figure 4b, Table 3). Essentially, if $\frac{K_r s_r}{a_r}$ is small, we can approximate NPP by $\frac{N_{in}}{l_{n,rate}(a_f n_{fl} + a_w n_w + a_r n_r)}$, which shares a similar structure to the baseline and fixed uptake coefficient models (Eq. 20, Eq. 17, and Eq. 10). Furthermore, Eq. 20 also depicts that increase in $a_r$ should lead to higher NPP and increase in $s_r$

or $K_r$ should lead to decreased NPP. However, these predictions depend on assumptions of $l_{n,rate}$ and $n_f$. If $l_{n,rate}$ or $n_f$ is small, NPP would be relatively less sensitive to $a_r$, $K_r$ or $s_r$.

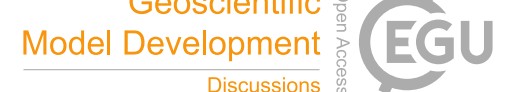



By comparison, representing N uptake as a saturating function of mineral N (Ghimire et al., 2016; Zaehle and Friend, 2010) no longer involves the VL-term nutrient recycling constraint on production (Figure 4c), which is predicted by Eq. 23. Actual VL-term NPP is determined only by $n_f$ along the photosynthetic constraint, meaning that the full $CO_2$ fertilization effect on production is realized with the increase in

$CO_2$. The magnitudes of the $CO_2$ fertilization effect at other time steps are comparable to those of the baseline model (Table 3), because the $N_{loss}$ term is smaller than $N_w$, $N_{Sp}$ or $N_{Ss}$ terms, meaning it has a relatively smaller effect on NPP at equilibrium. However, steeper nutrient recycling constraint curves are observed (Figure 4c), indicating a stronger sensitivity of the NPP response to changes in $n_f$.

**3.3 Potential NPP**

In several vegetation models, including CLM-CN, CABLE and JSBACH, potential (non-nutrient limited) NPP is calculated from light, temperature and water limitations. Actual NPP is then calculate by down-regulating the potential NPP to match nutrient supply. Here we term this the "potential NPP" approach. We examine this assumption in the QE framework following the implementation of this approach adopted

in CLM-CN (Bonan and Levis, 2010; Thornton et al., 2007). The potential NPP is reduced if mineral N availability cannot match the demand from plant growth:

$$P_{dem} = NPP_{pot}(a_f n_{fl} + a_w n_w + a_r n_r) \qquad \text{(Eq. 24)}$$

where $P_{dem}$ is the plant N demand, and $NPP_{pot}$ the potential NPP of the plant. Writing $(a_f n_f + a_w n_w + a_r n_r)$ as $n_{plant}$, the whole-plant N:C ratio, and the whole-soil N:C ratio as $n_{soil}$, we can calculate the immobilization N demand as:

$$I_{dem} = f C_{lit} s_t (n_{soil} - n_{plant}) \qquad \text{(Eq. 25)}$$

where $f$ is the fraction of litter C that becomes soil C, $C_{lit}$ is the total litter C pool, and $s_t$ is the turnover time of the litter pool. Actual plant N uptake is expressed as:



$$P_{act} = \min \left( \frac{N_{min} \, P_{dem}}{I_{dem} + P_{dem}}, P_{dem} \right) \qquad \text{(Eq. 26)}$$

Actual NPP is expressed as:

$$NPP_{act} = NPP_{pot} \frac{P_{act}}{P_{dem}} \qquad \text{(Eq. 27)}$$

For the VL constraint, we have $N_{in} = N_{loss}$. We can calculate $NPP_{pot}$ as:

$$NPP_{pot} = \frac{N_{in} \, (1 - l_n)}{l_n n_{plant}} \qquad \text{(Eq. 28)}$$

For an actual NPP, we need to consider the immobilization demand. Re-arranging the above, we get:

$$NPP_{act} = \frac{N_{in} \, (1 - l_n)}{l_n [n_{plant} + f(n_{soil} - n_{plant})]} \qquad \text{(Eq. 29)}$$

This equation removes the $NPP_{act}$ dependence on $NPP_{pot}$. It can be shown that the fraction of

$P_{dem}/(I_{dem} + P_{dem})$ depends only on the N:C ratios and $f$, not on $NPP_{pot}$. This means that there will be no eCO$_2$ effect on $NPP_{act}$.

As shown in Figure 5a, the potential NPP approach results in relatively flat nutrient recycling constraint curves, suggesting that the CO$_2$ fertilization effect is only weakly influenced by soil N availability. Despite a sharp instantaneous NPP response, CO$_2$ fertilization effects on *NPP*$_{act}$ are small at the M-, L-

and VL-term timescales (Table 3). This outcome can be understood from the governing equation for the nutrient recycling constraint, which removes *NPP*$_{act}$ dependence on *NPP*$_{pot}$ (Eq. 29). Although in the first instance, the plant can increase its production, over time the litter pool increases in size proportion to $NPP_{pot}$, meaning that immobilisation demand increases to match the increased plant demand, which leads to no overall change in the relative demands from the plant and the litter. This pattern is similar under

alternative wood N:C ratio assumptions (Figure 5b, Table 3).





### 3.4 Root exudation to prime N mineralisation

The priming effect is described as the stimulation of the decomposition of native soil organic matter, caused by larger soil carbon input under $eCO_2$ (van Groenigen et al., 2014). Experimental studies suggest that this phenomenon is widespread and persistent (Dijkstra and Cheng, 2007), but this process has not
been incorporated by most land surface models (Walker et al., 2015). Here we introduce a novel framework to induce priming effect on soil decomposition, and test its effect on plant production response to $eCO_2$ within the QE framework.

To account for the effect of priming on decomposition of SOM, we first introduce a coefficient to determine the fraction of root growth allocated to exudates, $a_{rhizo}$. Here we assumed that N:C ratio of the
rhizodeposition is the same as the root N:C ratio. The coefficient $a_{rhizo}$ is estimated by a function dependent on foliar N:C:

$$a_{rhizo} = a_0 + a_1 \cdot \frac{1/n_f - 1/n_{ref}}{1/n_{ref}} \qquad \text{(Eq. 30)}$$

where $n_{ref}$ is a reference foliar N:C ratio to induce plant N stress (0.04), and $a_0$ and $a_1$ are tuning coefficients (0.01 and 1, respectively). Within the QE framework, for the VL soil constraint we now have:

$$NPP = \frac{N_{in}}{[a_f n_{fl} + a_w n_w + a_r a_{rhizo} n_r + a_r (1 - a_{rhizo}) n_r]} \frac{l_n}{1 - l_n} \qquad \text{(Eq. 31)}$$

To introduce an effect of root exudation on the turnover rate of slow SOM pool, rhizodeposition is
transferred into the active SOM pool according to a microbial use efficiency parameter ($f_{cue,rhizo} = 0.3$). The extra allocation of NPP into the active SOM is therefore:

$$C_{rhizo} = NPP \cdot a_r \cdot a_{rhizo} \cdot f_{cue,rhizo} \qquad \text{(Eq. 32)}$$

The increased active SOM pool N demand is associated with the degradation rate of the slow SOM pool, expressed as:



$$k_{slow,new} = k_{slow} \cdot (1 + k_m) \cdot \frac{C_{rhizo}}{C_{rhizo} + k_m} \qquad \text{(Eq. 33)}$$

where $k_{slow}$ is the original decomposition rate of the slow SOM pool, and $k_m$ is a sensitivity parameter. The decomposition rate of the slow SOM pool affects $N_{Rs}$, the amount of N released from the slow SOM pools, as:

$$N_{Rs} = k_{slow,new} C_s [n_s (1 - \Omega_{ss}) - n_p \Omega_{ps}] \qquad \text{(Eq. 34)}$$

where $C_s$ is the slow SOM pool, and $\Omega_{ss}$ and $\Omega_{ps}$ are the proportion of C released through decomposition
of slow and passive SOM pools that subsequently enters slow SOM pool, respectively.

Root exudation and the associated priming effect results in a strong M-term plant response to $eCO_2$ when compared to the baseline model (Figure 6a in comparison to Figure 4a). In fact, the magnitude of the priming effect on M-term NPP response to $eCO_2$ is comparable to its L- and VL-term NPP responses, indicating a persistent $eCO_2$ effect over time (Table 3). A faster decomposition rate and therefore a smaller
pool size of the slow SOM pool are observed (Table 5). With a fixed wood N:C ratio assumption, NPP response to $eCO_2$ is drastically reduced at the M-term as compared to the model with a variable wood N:C assumption (Figure 6b), but is comparable to its corresponding baseline fixed wood N:C model (Table 3). Varying parameter coefficients ($a_0$, $a_1$, $f_{cue,rhizo}$ and $k_m$) affects the decomposition rates of slow soil organic pool and hence could lead to variation of the priming effect on M-term $CO_2$ response (Figure
S3). Further experimental studies are needed to better constrain these parameters. Adding root exudation without influencing slow SOM pool decomposition rate (Eq. 33) leads to a smaller predicted M-term $CO_2$ response than the model with the direct effect on the slow SOM pool. However, it also leads to a higher predicted M-term $CO_2$ response than the baseline model (Figure 7), because $a_r$ and $n_r$ affect the reburial fraction of the slow SOM pool, as shown in McMurtrie et al. (2000). Finally, the model with a variable
wood N:C assumption indicates that there is no increase in NUE (Table 2) at the M-term as compared to its L- and VL-term responses (Figure 6c). In comparison, the fixed wood N:C ratio assumption means



that there is a decreased wood "quality" (reflected via decreased N:C ratio), and therefore faster decomposition of slow SOM pool does not release much extra N to support the M-term $CO_2$ response, leading to a significant rise of NUE at the M-term (Figure 6d).

## 4 Discussion

### 4.1 Influence of alternative N uptake assumptions on predicted $CO_2$ fertilization

The QE analysis of the time-varying plant response to $eCO_2$ provides a quantitative framework to understand the relative contributions of different model assumptions governing the supply of N to plants in determining the magnitude of the $CO_2$ fertilization effect. Here, we evaluated how plant responses to $eCO_2$ are affected by widely used model assumptions relating to plant N uptake, soil decomposition, and immobilization demand under alternative wood N-C coupling strategies (variable and fixed wood N:C ratios). These assumptions have been adopted in land surface models such as O-CN (Zaehle and Friend, 2010), CABLE (Wang et al., 2007), LPJ-Guess N (Wårlind et al., 2014), JASBACH-CNP (Goll et al., 2012), ORCHIDEE-CNP (Goll et al., 2017a), and CLM4 (Thornton et al., 2007). In line with previous findings (Comins and McMurtrie, 1993; Dewar and McMurtrie, 1996; Kirschbaum et al., 1998; McMurtrie and Comins, 1996; Medlyn and Dewar, 1996), our results show that assumptions related to wood stoichiometry have a very large impact on estimates of plant responses to $eCO_2$. More specifically, models incorporating a fixed wood N:C ratio consistently predicted smaller $CO_2$ fertilization effects on production than models using a variable N:C ratio assumption (Table 3). Examples of models assuming constant (Thornton et al., 2007; Weng and Luo, 2008) and variable (Zaehle and Friend, 2010) plant tissue stoichiometry are both evident in the literature, and therefore, assuming all other model structure and assumptions are similar, prediction differences could potentially be attributed to the tissue stoichiometric assumption incorporated into these models, as suggested in some previous simulation studies (Medlyn et al., 2016; Medlyn et al., 2015; Meyerholt and Zaehle, 2015; Zaehle et al., 2014). Together with more appropriate representation of the trade-offs governing tissue C-N coupling (Medlyn et al., 2015), further





tissue biochemistry data is necessary to constrain this fundamental aspect of ecosystem model uncertainty (Thomas et al., 2015).

C-N coupled simulation models generally predict that the $CO_2$ fertilization effect on plant production is progressively constrained by soil N availability over time: the progressive nitrogen limitation hypothesis

(Luo et al., 2004; Norby et al., 2010; Zaehle et al., 2014). Here we showed similar temporal patterns in a model with different plant N uptake assumptions (Figure 4) and the relative demand assumption (Figure 5). In particular, the progressive N limitation effect on NPP is shown as a down-regulated M-term $CO_2$ response after the sharp instantaneous $CO_2$ fertilization effect on production is realized. However, the model incorporating a priming effect of C on soil N availability with a flexible wood N:C ratio assumption

induced a strong M-term $CO_2$ response (13% increase in NPP), thereby introducing a persistent $CO_2$ effect over time (Figure 6a). This strong M-term $CO_2$ response is due to an enhanced decomposition rate of soil organic matter, consistent with a series of recent observations and modelling studies (Finzi et al., 2015; Guenet et al., 2018; Sulman et al., 2014; van Groenigen et al., 2014). However, as a previous QE study showed, a significant increase in the M-term $CO_2$ response can occur via changes in litter quality into

slow SOM pool or increased N input into the system (McMurtrie et al., 2000). Our study differs from McMurtrie et al. (2000) in that we introduced an explicit effect of C priming on $k_{slow}$ – the decomposition rate of slow SOM pool – via extra rhizodeposition (Eq. 33). As such, a faster decomposition rate of slow SOM is observed (Table 5), equivalent to adding extra N for mineralization to support the M-term $CO_2$ response (Figure 6c). More complex models for N uptake, incorporating a carbon cost for nitrogen

acquisition, are being proposed (Fisher et al., 2010; Ghimire et al., 2016; Shi et al., 2015a); we suggest that the likely effects of introducing these complex sets of assumptions into large-scale models could usefully be explored with the QE framework.

A strong M-term and persistent $CO_2$ fertilization effects over time was also found by some models in Walker et al. (2015), but without introducing a priming effect. In models such as CLM, N losses from the

system are concentration dependent, and plant N uptake is a function of both N supply and plant demand. Increased plant N demand in models where N uptake is a function of plant N demand, reduces soil solution

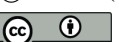


N concentration and therefore system N losses. This means that over time N can accumulate in the system in response to $eCO_2$ and sustain an $eCO_2$ response. Here, our QE framework considers N lost as a fixed rate that depends linearly on the mineral N pool, and the mineral N pool changes at different equilibrium time points. For example, as shown in Table S1, M-term N loss rate is significantly reduced under $eCO_2$

as compared to the VL-term N lost rate under $aCO_2$. This suggests a positive relationship between N lost and NPP, as embedded in Eq. 16.

We also showed that the magnitude of the $CO_2$ fertilization effect is significantly reduced at all time-scales when models incorporate the relative demand (or potential NPP) approach (Figure 5). Among all model assumptions tested, the relative demand approach induced the smallest M- to VL-term responses

(Table 3). It can be shown from equation derivation (Eq. 29) that the fraction $P_{dem}/(P_{dem} + I_{dem})$ depends only on the N:C ratios and $f$ (fraction of litter C become soil C), implying that models incorporating the relative demand assumption should show no response of NPP to $CO_2$. Both our study and simulation-based studies showed small $CO_2$ responses (Walker et al., 2015; Zaehle et al., 2014), possibly because the timing of $P_{dem}$ and $I_{dem}$ differs due to the fluctuating nature of GPP and N

mineralization at daily to seasonal time steps, such that N is limiting at certain times of the year but not at others. Additionally, models such as CLM have volatilization losses (not leaching) that are reduced under $eCO_2$, which may lead to production not limited by N availability, meaning that full $CO_2$ fertilization effect may be realized. Finally, leaching is simplified here, treated as a fixed fraction of the mineral N pool.  In models such as CLM or JASBACH, it is a function of soil soluble N concentration,

implying a dependency on litter quality (Zaehle et al., 2014).

### 4.2 Implications for probing model behaviours

Model-data intercomparisons have been shown as a viable means to investigate how and why models differ in their predicted response to $eCO_2$ (De Kauwe et al., 2014; Walker et al., 2015; Zaehle et al., 2014).

Models make different predictions because they have different model structures (Lombardozzi et al.,





2015; Meyerholt et al., 2016; Shi et al., 2018; Xia et al., 2013; Zhou et al., 2018), parameter uncertainties (Dietze et al., 2014; Wang et al., 2011), response mechanisms (Medlyn et al., 2015), and numerical implementations (Rogers et al., 2016). It is increasingly difficult to diagnose model behaviours from the multitude of model assumptions incorporated into the model. Furthermore, while it is true that the models

can be tuned to match observations within the domain of calibration, models may make correct predictions but based on incorrect or simplified assumptions (Medlyn et al., 2005; Medlyn et al., 2015; Walker et al., 2015). As such, diagnosing model behaviours can be a challenging task in complex plant-soil models. In this study, we showed that the effect of a model assumption on plant response to $eCO_2$ can be analytically predicted by solving together the photosynthetic and nutrient recycling constraints. This provides a

constrained model framework to evaluate the effect of individual model assumptions without having to run a full set of sensitivity analyses, thereby providing *a priori* understanding of the underlying response mechanisms through which the effect is realized. We suggest that before implementing a new function into the full structure of a plant-soil model, one could use the QE framework as a testbed to examine the effect of the new assumption.

The QE framework requires that additional model assumptions be analytically solvable, which is increasingly not the case for complex modelling structures. However, as we demonstrate here, studying the behaviour of a reduced-complexity model can nonetheless provide real insight into model behaviour. In some cases, the QE framework can highlight where additional complexity is not valuable. For example, here we showed that adding complexity in the representation of plant N uptake did not result in

significantly different predictions of plant response to $eCO_2$. Where the QE framework indicates little effect of more complex assumptions, there is a strong case for keeping simpler assumptions in the model. However, we do acknowledge that the QE framework operates on time-scales of > 5 years; where fine-scale temporal responses are important, the additional complexity may be warranted.

A related model assumption evaluation tool is the traceability framework, which decomposes complex

models to various simplified component variables such as ecosystem C storage capacity or residence time, and hence helps to identify structures and parameters that are uncertain among models (Shi et al., 2015b;



Xia et al., 2013; Xia et al., 2012). Both the traceability and QE frameworks provide analytical solutions to describe how and why model predictions diverge. The traceability framework decomposes complex simulations into a common set of component variables, explaining differences due to these variables. In contrast, the QE analysis investigates the impacts and behaviour of a specific model assumption, which

is more indicative of mechanisms and processes. Subsequently, one can relate the effect of a model assumption more mechanistically to the processes that govern the relationship between plant N:C ratio and NPP, as depicted in Figure 1, thereby facilitating the efforts to reduce model uncertainties.

Models diverge in future projections of plant responses to increases in $CO_2$ because of the different assumptions that they make. Applying model evaluation frameworks, such as the QE framework, to

attribute these differences will not necessarily reduce multi-model prediction spread in the short-term (Lovenduski and Bonan, 2017). Many model assumptions are still empirically derived, and there is a lack of mechanistic and observational constraints on the effect size, meaning that it is important to apply models incorporating diverse process representations. However, use of the QE framework can provide crucial insights into why model predictions differ, and thus help identify the critical measurements that

would allow to discriminate among alternative models. As such, it is an invaluable tool for model inter-comparison and benchmarking analysis. We recommend use of this framework to analyze likely outcomes of new model assumptions before introducing them to complex model structures.

**Code availability**

Code repository is publicly available via GitHub (https://github.com/mingkaijiang/QEframework.git).





**Author contribution**

BEM and MJ designed the study; MJ, BEM and SZ performed the analyses; APW, MDK and SZ designed the priming effect equations; all authors contributed to result interpretation and manuscript writing.

5  **Competing interests**

Authors declare no competing interests.

**Acknowledgements**

SZ and SC were supported by the European Research Council (ERC) under the European Union's
10  Horizon 2020 research and innovation programme (QUINCY; grant no. 647204) and the German Academic Exchange Service (DAAD; project-id 57318796). DE and MJ were also supported by the DAAD.



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





Table 1: A brief summary of the processes and model assumptions evaluated based on the quasi-equilibrium analyses

| Processes | Assumptions | Findings | Key reference |
|---|---|---|---|
| Stoichiometry | Wood N:C flexibility | Flexible wood N:C ratio induced a strong NPP response to $eCO_2$. | Comins and McMurtrie, 1993; Comins, 1994; Kirschbaum et al., 1994; McMurtrie and Comins, 1996; Kirschbaum et al., 1998 |
| | Soil N:C flexibility | Soil N:C ratio flexibility fundamentally underpin NPP response to $eCO_2$. | McMurtrie and Comins, 1996; Medlyn et al., 2000; McMurtrie et al., 2001 |
| | Litter NC flexibility | Decreased new litter N:C ratio did not significantly alter NPP response to $eCO_2$, but a substantial decrease in old litter N:C ratio led to a significant $CO_2$ effect at the medium-term. | McMurtrie et al., 2000 |
| Allocation | Dynamic allocation as a response to changes in leaf N:C ratio | Changes in C allocation between different parts do not significantly alter NPP response to $eCO_2$. | Kirschbaum et al., 1994 |
| | Linear stem and leaf allocation coupling | With stem allocation proportional to leaf allocation, NPP response to $eCO_2$ is significant, even when N deposition is unchanged. | Medlyn and Dewar, 1996 |
| Nutrient supply and loss | N fixation | N deficit induced by $CO_2$ fertilization can be eliminated by stimulation of N fixation. | Comins, 1994 |
| | N fixation | Enhanced N fixation via root exudation leads to a small effect on production in the short term but a very large effect in the long term. | McMurtrie et al., 2000 |
| | Leaf N retranslocation | Changes in leaf N retranslocation fraction do not significantly affect NPP response to $eCO_2$. | Kirschbaum et al., 1994 |
| | Litter supply | Increased litter quantity only leads to a minimal $CO_2$ effect on production. | McMurtrie et al., 2000 |
| | Nutrient supply and loss | Systems that are more open with respect to nutrient gains and losses are likely to be more responsive to $eCO_2$. | Kirschbaum et al., 1998 |
| | N mineralization | Increased temperature induced a long-term increase in NPP response to $eCO_2$ because of increased N mineralization and plant N uptake rates | Medlyn et al., 2000 |
| | N immobilization | When both T and $CO_2$ increase, C sink is insensitive to variability in soil N:C ratio, however, with fixed soil N:C, C sink is primarily a temperature response, whereas with variable soil N:C, it is a combined temperature-$CO_2$ response. | McMurtrie e a., 2001 |
| Photosynthesis | LUE coefficient | Effect of leaf N:C ratio on LUE coefficient induces a small effect on $CO_2$ sensitivity of plant. | Kirschbaum et al., 1994 |
| | SLA | Introducing leaf N:C dependency of SLA induces no significantly different NPP response to $eCO_2$. | Kirschbaum et al., 1994 |





Table 2: Definitions of key variables for the baseline equations

| Symbol | Definition | Value | Unit |
|---|---|---|---|
| $aCO_2$, $eCO_2$ | Ambient and elevated $CO_2$ concentration, respectively | 400, 800 | ppm |
| $N_{in}$ | Total nitrogen into the system (atmospheric deposition and fixation) | 0.004 | t ha$^{-1}$ yr$^{-1}$ |
| $T_{air}$, $T_{soil}$, $T_{leaf}$ | Temperature of air, soil, and leaf, respectively | 20, 15, 25 | °C |
| CUE | Plant carbon use efficiency | 0.5 | unitless |
| NUE | Plant nitrogen use efficiency = NPP / $N_u$ | Calculated | kg C kg N$^{-1}$ |
| $\sigma$ | Specific leaf area | 5 | m$^2$ kg$^{-1}$ |
| $\omega$ | Carbon content of biomass | 0.45 | unitless |
| $a_f$, $a_r$, $a_w$ | Carbon allocation fraction to leaf, root and wood, respectively | 0.2, 0.2, 0.6 | unitless |
| $n_f$, $n_r$, $n_w$, $n_{fl}$ | N:C ratio of leaf, root, wood, and leaf litter, respectively | | unitless |
| $t_f$ | Leaf retranslocation rate | 0.5 | yr$^{-1}$ |
| $r_w$, $r_r$ | Proportion of wood and root N:C ratio to leaf N:C ratio, respectively | 0.005, 0.7 | unitless |
| $s_f$, $s_r$, $s_w$ | Turnover rates of leaf, root and wood, respectively | 0.5, 1.5, 0.01 | yr$^{-1}$ |
| $n_a$, $n_s$, $n_p$ | C:N ratio for active, slow, passive SOM pool, respectively | 15, 20, 10 | unitless |
| $l_n$ | Fraction of N mineralization lost from the system | 0.05 | unitless |
| $l_{n,\ rate}$ | Mineral N pool lost rate | 0.05 | yr$^{-1}$ |
| $O_{acq}$, $O_{resorb}$, $O_{active}$ | Total, resorption, and active C cost of N acquisition, respectively | Calculated | kg C kg N$^{-1}$ |
| $\Omega_{sf}$, $\Omega_{pf}$ | Proportion of leaf litter enters into slow and passive SOM pool, respectively | Calculated | unitless |
| $\Omega_{sr}$, $\Omega_{pr}$ | Proportion of root litter enters into slow and passive SOM pool, respectively | Calculated | unitless |
| $\Omega_{sw}$, $\Omega_{pw}$ | Proportion of wood litter enters into slow and passive SOM pool, respectively | Calculated | unitless |
| $N_{Ss}$, $N_{Sp}$, $N_{Sw}$ | N stored in slow, passive SOM, and wood pool, respectively | Calculated | t ha$^{-1}$ yr$^{-1}$ |
| $N_{Rs}$, $N_{Rp}$, $N_{Rw}$ | N released from slow, passive SOM, and wood pool, respectively | Calculated | t ha$^{-1}$ yr$^{-1}$ |
| $N_U$ | N uptake rate | Calculated | t ha$^{-1}$ yr$^{-1}$ |
| $N_{min}$ | Mineral N pool | Calculated | t ha$^{-1}$ |



Table 3: Magnitudes of the $CO_2$ fertilization effect on net primary production (NPP) at various time steps for different model assumptions. $NPP_a$ and $NPP_e$ represent very long-term equilibrium point of NPP at ambient and elevated $CO_2$ conditions, respectively. I, M, L, and VL represent percent change in NPP as a result of elevated $CO_2$ at instantaneous, medium, long, and very-long term time points, respectively. All
5   experiments except "baseline, fixed wood NC" assume variable wood N:C ratio.

| Experiment | $NPP_a$ | $NPP_e$ | I | M | L | VL |
|---|---|---|---|---|---|---|
| **Baseline model, variable wood NC** | 1.67 | 1.90 | 15.1 | 3.2 | 12.3 | 13.3 |
| **Baseline model, fixed wood NC** | 1.49 | 1.66 | 15.9 | 0.8 | 7.9 | 10.9 |
| **Explicit N uptake, fixed coefficient, variable wood NC** | 1.68 | 1.91 | 15.1 | 3.2 | 12.4 | 13.3 |
| **Explicit N uptake, fixed coefficient, fixed wood NC** | 1.52 | 1.68 | 15.8 | 0.8 | 8.2 | 11.1 |
| **Explicit N uptake, saturating function of root, variable wood NC** | 1.68 | 1.91 | 15.1 | 3.2 | 12.4 | 13.3 |
| **Explicit N uptake, saturating function of Nmin, variable wood NC** | 1.71 | 1.96 | 15.0 | 3.2 | 13.7 | 15.0 |
| **Priming, variable wood NC** | 1.67 | 1.90 | 15.1 | 12.2 | 12.0 | 13.3 |
| **Priming, fixed wood NC** | 1.49 | 1.66 | 15.9 | 1.8 | 8.3 | 10.9 |
| **Relative demand, variable wood NC** | 1.35 | 1.42 | 16.6 | 0.3 | 2.9 | 4.9 |
| **Relative demand, fixed wood NC** | 1.13 | 1.15 | 17.9 | 0.2 | 1.1 | 1.7 |



Table 4. Relationship between nitrogen uptake coefficient (u) and quasi-equilibrium points of leaf N:C ratio ($n_f$) and net primary production (NPP) at the very-long (VL), long (L), medium (M) and instantaneous time points.

| $u$ (yr⁻¹) | $CO_2$ (ppm) | $n_f$ | | | NPP (kg C m⁻² yr⁻¹) | | | |
|---|---|---|---|---|---|---|---|---|
| | | VL | L | M | VL | L | M | I |
| 0.2 | 400 | 0.0049 | 0.0049 | 0.0049 | 1.35 | 1.35 | 1.35 | - |
| 0.2 | 800 | 0.0043 | 0.0039 | 0.0026 | 1.53 | 1.51 | 1.39 | 1.57 |
| 0.5 | 400 | 0.01 | 0.01 | 0.0107 | 1.54 | 1.54 | 1.54 | - |
| 0.5 | 800 | 0.01 | 0.008 | 0.005 | 1.75 | 1.72 | 1.59 | 1.78 |
| 1 | 400 | 0.02 | 0.02 | 0.0196 | 1.68 | 1.68 | 1.68 | - |
| 1 | 800 | 0.017 | 0.016 | 0.0089 | 1.91 | 1.89 | 1.74 | 1.94 |
| 2 | 400 | 0.036 | 0.036 | 0.036 | 1.81 | 1.81 | 1.81 | - |
| 2 | 800 | 0.032 | 0.029 | 0.014 | 2.05 | 2.03 | 1.85 | 2.07 |
| 5 | 400 | 0.084 | 0.084 | 0.084 | 1.95 | 1.95 | 1.95 | - |
| 5 | 800 | 0.075 | 0.062 | 0.032 | 2.21 | 2.17 | 2.04 | 2.23 |





Table 5. Effect of priming on key soil process coefficients. Coefficient $k_{slow}$ is the decomposition coefficient for the slow SOM pool ($yr^{-1}$); apass is the reburial fraction of the passive SOM (i.e. the fraction of passive SOM re-enters passive SOM); $a_{slow}$ is the reburial fraction of the slow SOM; $\Omega_p$ is the burial coefficient for plant materials entering the passive SOM pool; $\Omega_s$ is the burial coefficient for plant materials entering the slow SOM pool; and $C_{slow}$ is the total carbon stock of the slow SOM pool (g C m$^{-2}$). Both models assume variable wood N:C ratio.

| Model | $k_{slow}$ | $a_{pass}$ | $a_{slow}$ | $\Omega_p$ | $\Omega_s$ | $C_{slow}$ |
|---|---|---|---|---|---|---|
| baseline | 0.067 | 0.011 | 0.211 | 0.002 | 0.155 | 4726 |
| priming | 0.185 | 0.011 | 0.211 | 0.001 | 0.163 | 1624 |





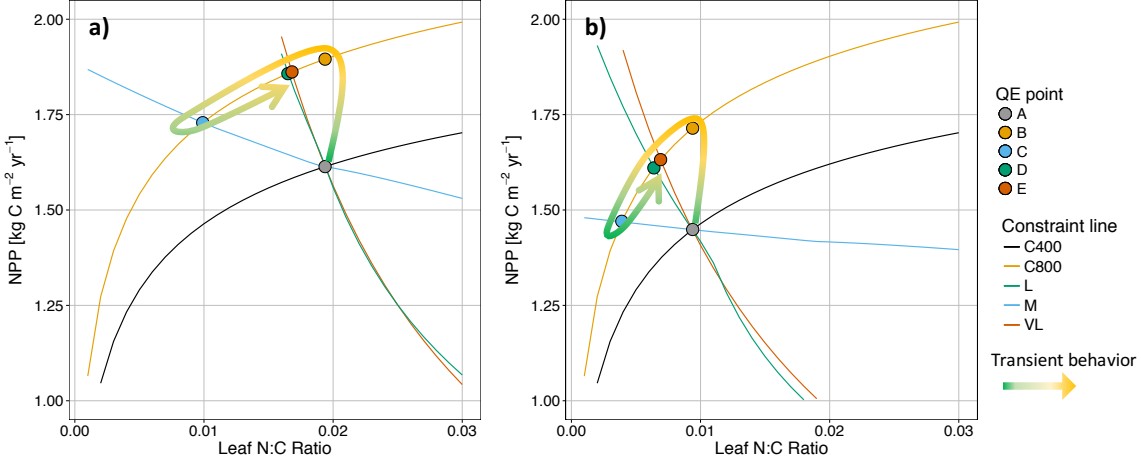

Figure 1: Graphic expression of the baseline quasi-equilibrium framework in understanding plant production response to elevated $CO_2$, based on photosynthetic (C400, C800 refer to $CO_2$ = 400 ppm and 800 ppm, respectively) and nitrogen cycling constraints at the medium (M), long (L) and very long (VL) terms, under the assumption of a) variable wood N:C ratio, and b) fixed wood N:C ratio. The photosynthetic constraint is an analytical expression of the Farquhar leaf photosynthesis model that relates leaf chemistry (i.e. NC ratio) with production, simplifying leaf to canopy scaling. The nutrient recycling constraint is an analytical expression of the soil nutrient down-regulation effect on production, assuming soil organic matter structures as in Figure 2. The quasi-equilibrium points at various timescales (A, C, D and E) were calculated by solving for the intersection of the photosynthetic and nutrient cycling constraints through the two-timing approximation. Initially the system is in equilibrium between photosynthetic N demand and soil N supply at $CO_2$ = 400 ppm (A). The instantaneous response to doubling of $CO_2$ is a sharp increase in production at a constant leaf N concentration (B). Under nutrient limited condition, soil N supply cannot sustain this increase in production over time. A negative feedback moves the quasi-equilibrium point towards point C, where the M-term pools equilibrate with $eCO_2$. The system gradually moves toward point D and E as the L and VL pools equilibrate. The downward slopes of the N recycling constraint curves with increasing leaf N:C ratio is due to the increased proportional loss of mineralized N through leaching as the rate of N cycling increases with leaf N concentration.



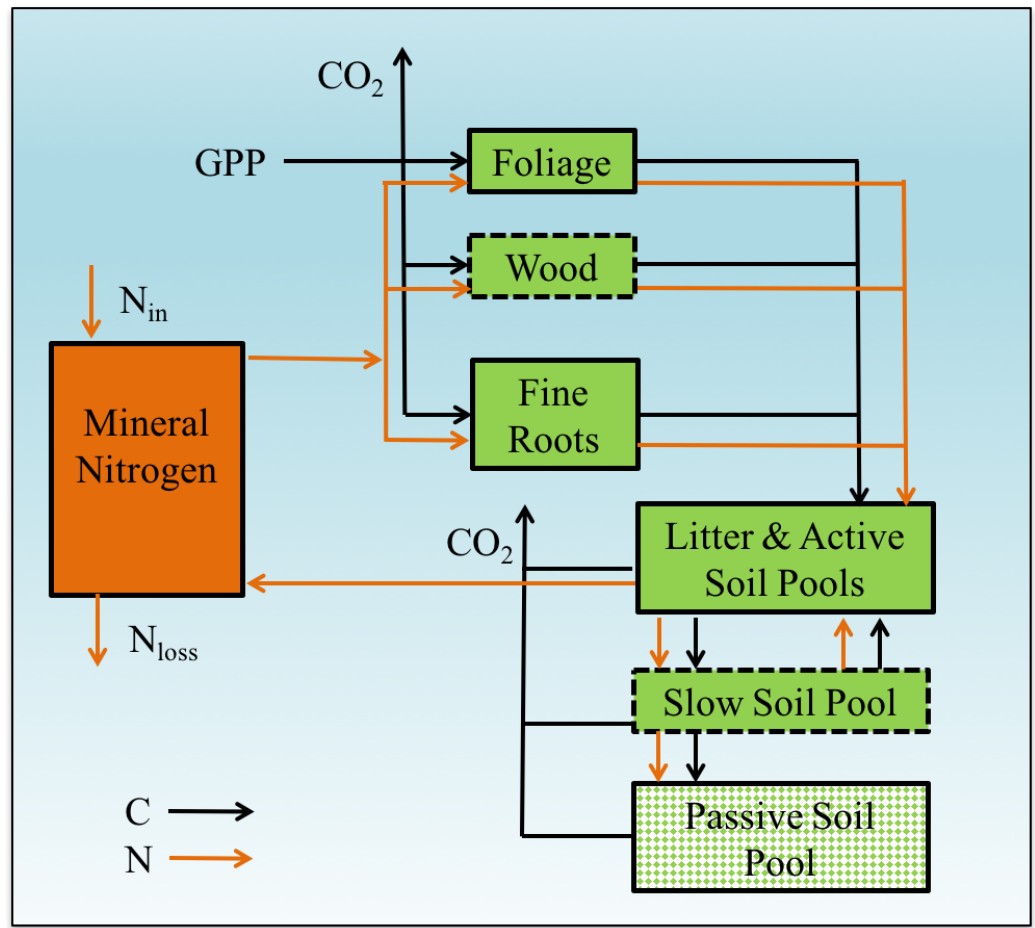

Figure 2: Framework of the Generic Decomposition And Yield (G'DAY) model. Boxes represent pools; arrowed line represent fluxes. Boxes with dotted boundaries are M term recycling pools (wood and slow soil). Box filled with diamonds is the L term recycling pool (passive soil).



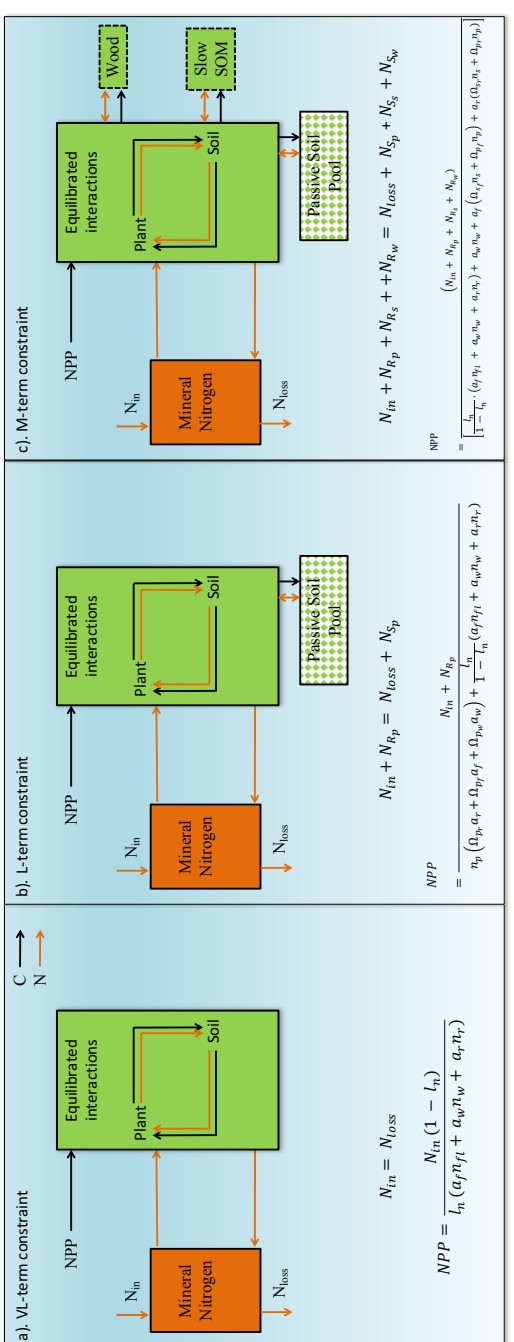

Figure 3: Graphic and mathematical illustrations of the a) very-long (VL) term, b) long (L) term, and c) medium (M) term nutrient recycling constraints. VL-constraint considers all plant-soil processes are in equilibrium, L-constraint considers all but passive SOM are in equilibrium, and M-constraint considers all but woody biomass, slow and passive SOM pools are in equilibrium.





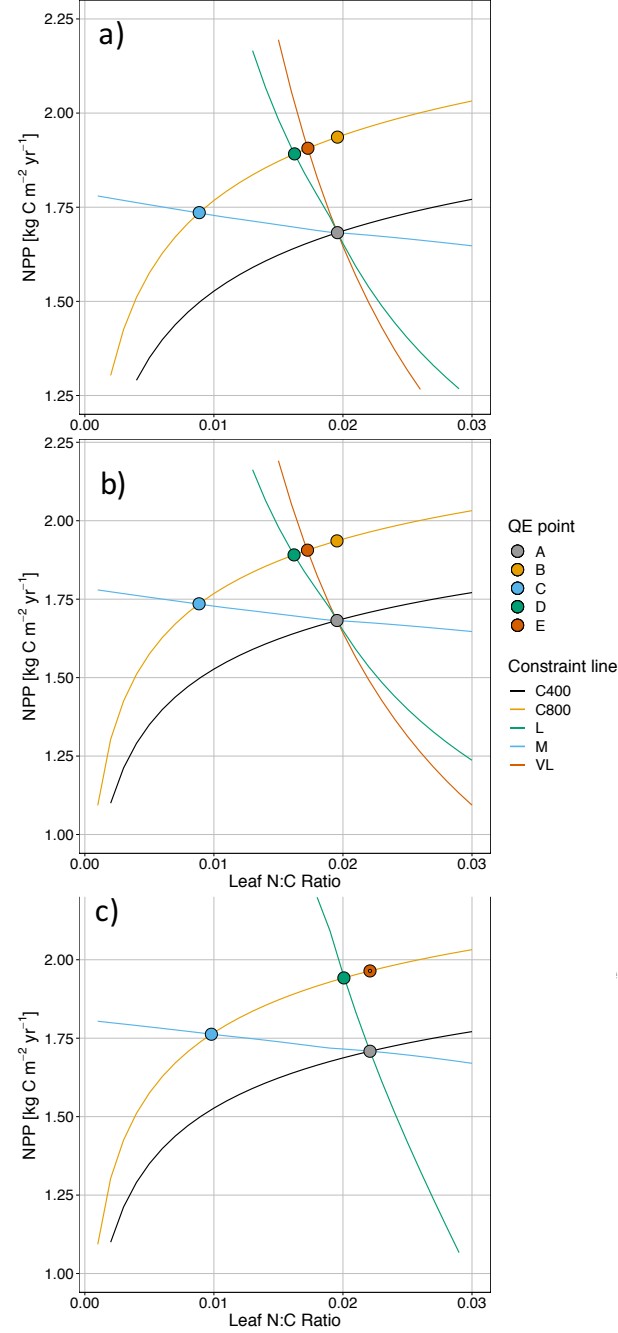



Figure 4: Graphic interpretation of the effect of different nutrient uptake assumptions on plant response to $CO_2$ fertilization. Functions are: a) plant N uptake as a function of a constant coefficient, with a variable wood N:C ratio assumption, b) plant N uptake as a saturating function of root biomass and also linearly depends upon mineral N pool, and c) plant N uptake as a saturating function of mineral N pool and also
5    linearly depends upon root biomass. Constraint lines C400, C800, M, L and VL refer to photosynthetic constraints at $CO_2$ = 400 ppm, $CO_2$ = 800 ppm, medium term, long term, and very-long term nutrient recycling constraints, respectively. Point A is the quasi-equilibrium point at $CO_2$ = 400 ppm, point B is the instantaneous response point at elevated CO2, points C, D, and E are the M, L and VL term equilibrium points at elevated $CO_2$. The N uptake coefficient is set to 1 $yr^{-1}$.





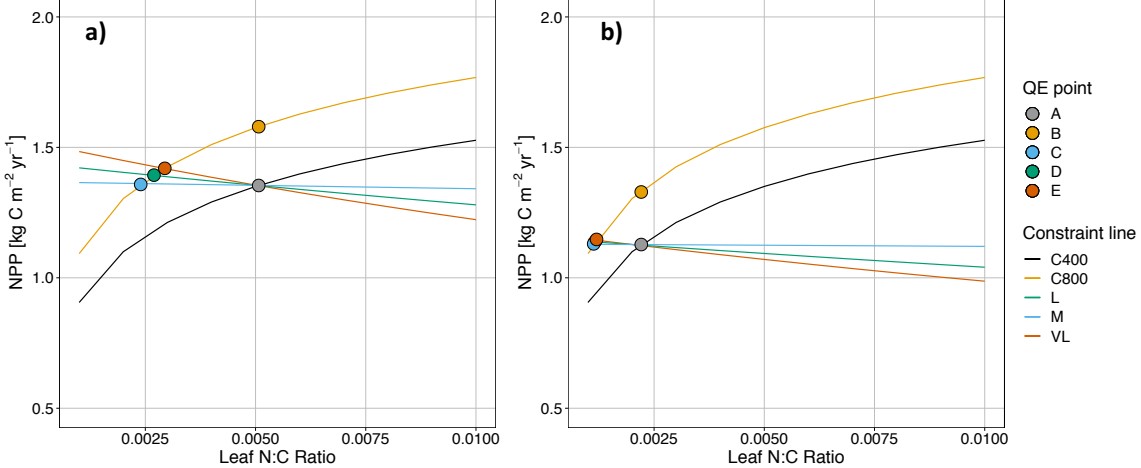

Figure 5: Graphic interpretation of the effect on $CO_2$ responses with models incorporating relative demand assumption, based on variable (a) and fixed (b) wood N:C ratio assumptions. Constraint lines C400, C800, M, L and VL refer to photosynthetic constraints at $CO_2$ = 400 ppm, $CO_2$ = 800 ppm, medium term, long term, and very-long term nutrient recycling constraints, respectively. Point A is the quasi-equilibrium point at $CO_2$ = 400 ppm, point B is the instantaneous response point at elevated $CO_2$, points C, D, and E are the M, L and VL term equilibrium points at elevated $CO_2$.

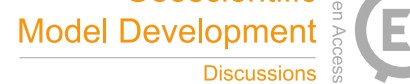



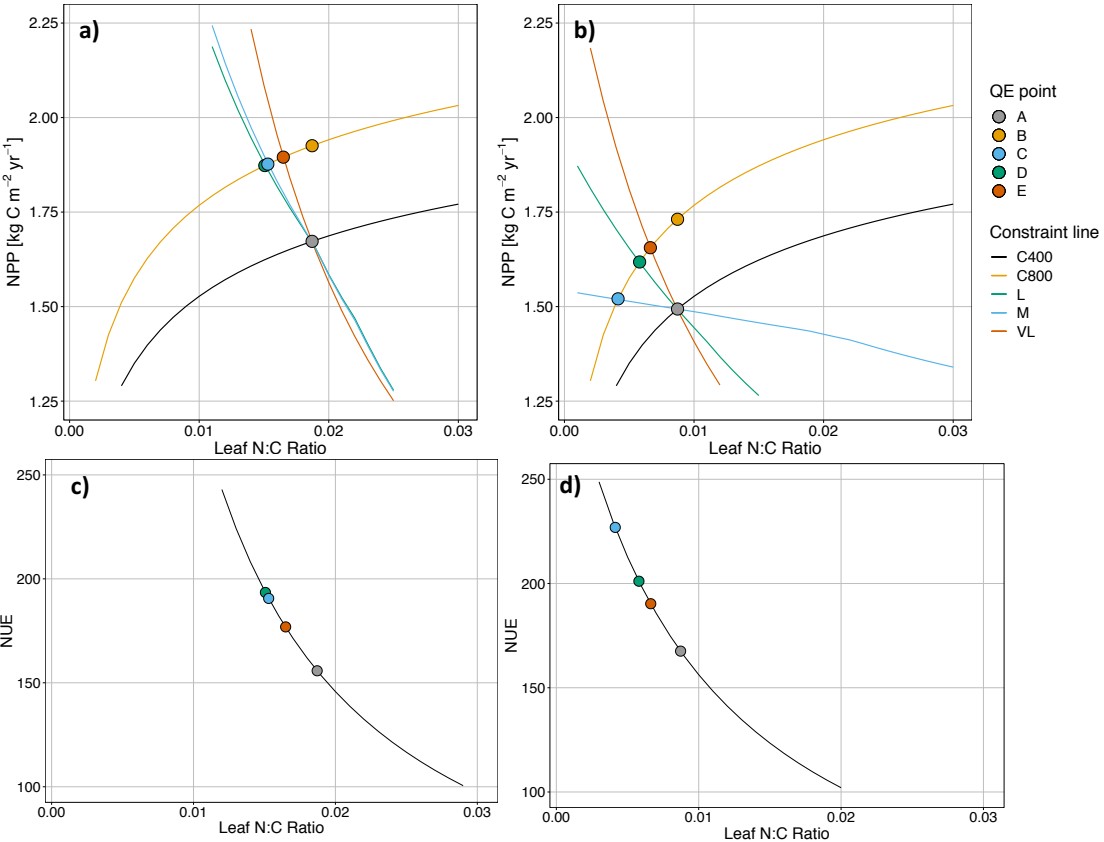

Figure 6: Graphic interpretation of the priming effect on plant net primary production (a and b) and nitrogen use efficiency (c and d) response to $CO_2$ fertilization, under variable wood N:C ratio (a and c) and fixed wood N:C ratio assumptions (b and d). Constraint lines C400, C800, M, L and VL refer to photosynthetic constraints at $CO_2$ = 400 ppm, $CO_2$ = 800 ppm, medium term, long term, and very-long term nutrient recycling constraints, respectively. Point A is the quasi-equilibrium point at $CO_2$ = 400 ppm, point B is the instantaneous response point at elevated $CO_2$, points C, D, and E are the M, L and VL term equilibrium points at elevated $CO_2$.



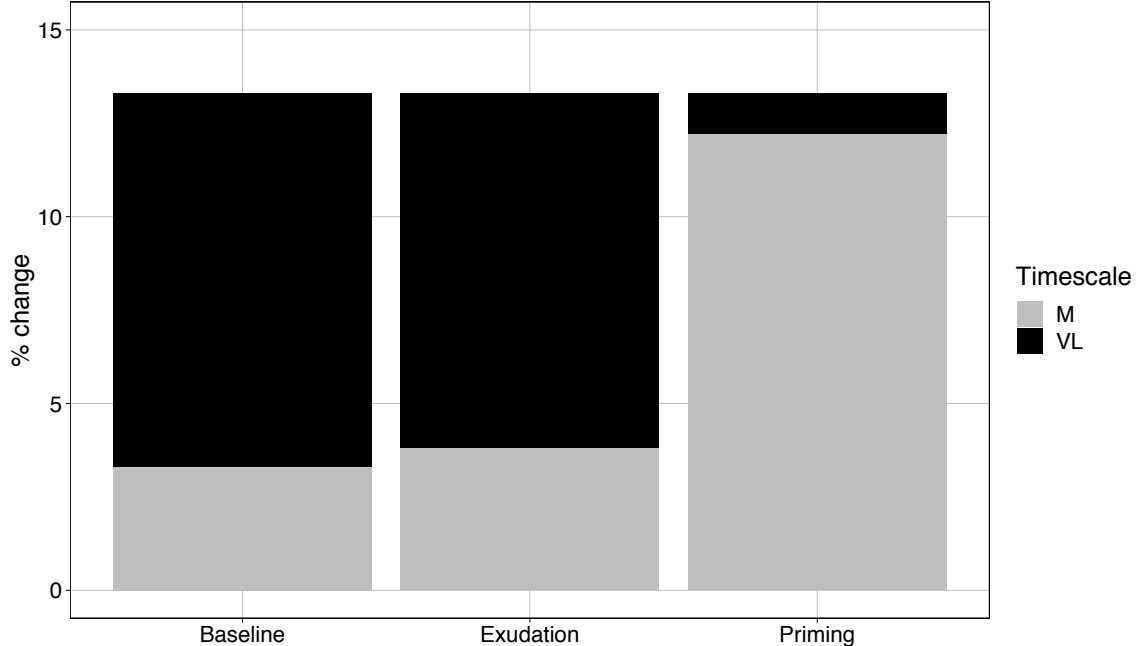

Figure 7: Comparison of medium term (M) and very long term (VL) net primary production response to elevated $CO_2$ (% change), with models incorporating no priming and exudation effect (baseline), only
5    exudation effect (exudation), and both exudation and priming effect (priming).