# Peer review of "The quasi-equilibrium framework re-visited: analyzing long-term CO2 enrichment responses in plant-soil models"

_Geoscientific Model Development, 2018_

## Short Comment (SC1) · 18 Dec 2018

Dear authors,

in my role as Executive editor of GMD, I would like to bring to your attention our Editorial version 1.1: http://www.geosci-model-dev.net/8/3487/2015/gmd-8-3487-2015.html This highlights some requirements of papers published in GMD, which is also available on the GMD website in the 'Manuscript Types' section: http://www.geoscientific-model-development.net/submission/manuscript_types.html

In particular, please note that for your paper, the following requirement has not been met in the Discussions paper:

- "The main paper must give the model name and version number (or other unique identifier) in the title."

Please provide explicitly the name (or its acronym) and the version number of the framework in the title of your revised manuscript.

GMD is encouraging authors to provide a persistent access to the exact version of the source code used for the model version presented in the paper. As explained in https://www.geoscientific-model-development.net/about/manuscript_types.html the preferred reference to this release is through the use of a DOI which then can be cited in the paper. For projects in GitHub a DOI for a released code version can easily be created using Zenodo, see https://guides.github.com/activities/citable-code/ for details. You may consider to upload the program code of the specifc version of the paper as a supplement or make the code and data of the exact model version described in the paper accessible through a DOI (digital object identifier). In case your institution does not provide the possibility to make electronic data accessible through a DOI you may consider other providers (eg. zenodo.org of CERN) to create a DOI. Please note that in the code accessibility section you can still point the reader to the GitHub repository for the newest version even if you use a DOI for the relevant releases.

Yours, Astrid Kerkweg
* * *

---

## Referee Comment (RC1) · Anonymous Referee #1 · 17 Jan 2019

Overall, this paper is excellent. It usefully adds to the body of experimental results based on the use of quasi-equilibrium models and further develops our understanding of the importance of key processes and assumptions.

Having said that, I was very happy with reading the paper up to page 19, line 5. Introduction and the associated discussion of past research was excellent, detailed and informative and structured to derived useful information and leading to new questions.

The description of the model was also clear and gave all the relevant model components. It was maybe a little long (34 equations), and there a danger that key model steps might have be swamped within a sea of less important ones. There might be a

point in moving some of the model detail to Supplemental Information, and only presenting the key steps in the main text.

However, the remainder of the paper was not well presented. Page 19, line 6 should have started a 'Results Section' where the various figure and tables could have been presented and discussed in some detail. As it is, all the key findings were dumped here within half a page. I do not regard this as satisfactory.

Each figure shows important information that is not immediately obvious. It needs careful text that explains to the reader what we can learn from each figure. A general 'data dump' with virtually no explanation is never a good way to proceed, but totally unacceptable in this case, as the essence of the modelling is not immediately obvious, but the reader needs to be led through the various figures to extract the key insights gained from each.

Some of that detail is then given in the Discussion, like page 21, line 5 onwards, but only very briefly. That reference is too brief on its own and would have needed a proper description in a results section that could then be referred to. So, the Discussion might be OK if it had an appropriate Results section. But without a Results section, the reference to the various figures is still too brief to be readily and fully understood by the reader.

So, all in all, I would regard the paper as not acceptable in its current form, but that is entirely due to the lacking Results Section and insufficient description of the modeled findings. If that can be added, and the Discussion section then be modified to appropriately refer to text in the Results Section, the paper should be able to make a really strong contribution to the literature.

Minor comments: Page 3, line 7: The authors introduce the abbreviations QE for 'quasi-equilibrium'. That is unnecessary in my view and just obscures the subsequent text. 'Quasi-equilibrium' is short enough and can continue to be used throughout the paper. No need to confuse the reader by an unnecessary abbreviation.

Page 7, line 7: When the authors mention 'concentration-carbon' feedback, I assume they mean 'CO2-carbon' feedback. It would be better if that could be spelled out more explicitly as 'CO2-carbon' or something else that would leave the reader in doubt as to what concentration is referred to.

Page 7, line 22: Here, it states that in assumption 3, N uptake is modeled as a saturating function of root biomass. This makes it sound as though there were no upper limit to N uptake other than that imposed by root biomass. However, the detailed model description states that N uptake is also dependent on mineralized N, which seems like a sensible assumption. Just make sure that in the initial description of this assumption, it is also made clear that mineralized N is a co-limiting factor. Currently, that is not included and gives a misleading impression of the model assumption.

Nothing to add to the Model Description. The text after that needs some bigger overhaul as mentioned above, and I have refrained from referring to specific details as they will hopefully be changed in a bigger re-write.

---

## Referee Comment (RC2) · Anonymous Referee #2 · 24 Jan 2019

This study tries to establish a quasi-equilibrium (QE) analytical framework, introduced by Comins & McMurtrie (1993), for evaluating model assumptions on carbon-nitrogen interaction in influencing ecosystem responses to elevated CO2. Overall, this paper is extremely valuable for understanding a variety of assumptions in influencing model outputs of carbon and nitrogen coupling.

I particularly like your examples on page 23 to make a point that "the QE framework can highlight where additional complexity is not valuable."

Here are a few suggestions to improve your manuscript:

First, the authors may consider improve the readability of your paper so that your message can go more miles. It is quite competent of the authors to work out all those equations in section 3. But those equations will hinder delivering your message as not all the ecologists or even modelers will go over those equations when they read your paper. In addition, would it be possible to convert Table 1 to a graph so that readers can quickly get your message? To me, Table 1 is probably the most important part of your manuscript. Even though I am familiar with the subject, it still takes me a while to go over the table. Converting it to a figure may help deliver your message faster. Moreover, the abstract I don't think deliver the message well, especially the second half.

Second, the work by Comins & McMurtrie (1993) is great. But, during the same period in 1990s, Dr. Edward Rastetter has developed the Multiple Element Limitation (MEL) model of carbon-nitrogen interactions. He published a few papers to illustrate similar principles on carbon-nitrogen interactions as revealed by G'DAY. In fact, Ed Rastetter also lumped all those assumptions (or processes) into three categories as in the first three items of your Table 1. MEL further shows the time scales at which each of the three categories of processes plays. In other words, MEL not only gives information about the equilibrium responses but also offers information about C/N interaction to influence transient dynamics. I think the authors at least should acknowledge Ed's work in your manuscript.

Third, it is fine that the G'DAY model offers an analytical framework to evaluate model assumptions on carbon and nitrogen interactions. However, the impacts (or sensitivity) of those assumptions evaluated by the framework depend on the ranges of the variables you changed. For example, your analysis shows that wood N:C flexibility is very important for modeling carbon and nitrogen interactions. What ranges of wood N:C did those studies change? Do those ranges realistically match observations? Lots of data are available to evaluate those ranges. In fact, several studies have evaluated the ranges of changes of those variables (e.g., Liang et al. 2016). Bringing observations into your study may require the authors to do additional work but will improve quality

of your study. At least the authors should add discussion on observed vs. modeled ranges of changes.

Forth, if the authors want to popularize the QE framework to be used by the broad community, they may develop a simpler scheme for others to use. The extensive list of those equations may make it very difficult for others to use.

Reference: Junyi Liang, Xuan Qi, Lara Souza, and Yiqi Luo. 2016. Processes regulating progressive nitrogen limitation under elevated carbon dioxide: a meta-analysis. Biogeosciences, 13, 2689-2699.

---

## Author Comment (AC2) · 14 Feb 2019

**Response to Reviewer #1**

Overall, this paper is excellent. It usefully adds to the body of experimental results based on the use of quasi-equilibrium models and further develops our understanding of the importance of key processes and assumptions.

Having said that, I was very happy with reading the paper up to page 19, line 5. Introduction and the associated discussion of past research was excellent, detailed and informative and structured to derived useful information and leading to new questions. The description of the model was also clear and gave all the relevant model components.
**Responses**: We thank the reviewer for this positive overall comment on our paper.

It was maybe a little long (34 equations), and there a danger that key model steps might have be swamped within a sea of less important ones. There might be a point in moving some of the model detail to Supplemental Information, and only presenting the key steps in the main text.
**Responses**: We agree that we may have included too many equations in our manuscript. In fact, equations governing the baseline model (Eq. 1-15) have been published previously by a number of insightful studies, which we have provided in-depth reviews in our literature review section (e.g. Comins and McMurtrie, 1993; Kirschbaum et al. 1994, etc.). However, to comply with journal requirement, we need to "contain the justification of the model structure in the main body of the paper" (stated by the chief editor prior to our submission). We therefore cannot move our baseline model derivations into the Supplementary Materials under this consideration. In our revision, we propose to move these baseline derivations, together with derivations of new model assumption, into an Appendix as an integral part of the manuscript. This will help smooth the reading of the story, at the same time providing all fundamental analytical details in the main text. We hope that the journal could allow us adding an Appendix as part of the main text.

However, the remainder of the paper was not well presented. Page 19, line 6 should have started a 'Results Section' where the various figure and tables could have been presented and discussed in some detail. As it is, all the key findings were dumped here within half a page. I do not regard this as satisfactory.
**Responses**: We must respectfully point out that the reviewer may have misinterpreted our original Method and Result section. It was our deliberate intention to keep the Method and Result section together because the quasi-equilibrium framework is an analytical way of interpreting the likely impact of a model assumption to model behaviors. Therefore, derivations of equations are fundamental and deeply linked to the analysis of results. In our Method and Result section, we firstly described the baseline quasi-equilibrium framework (section 3.1), then assessed several recently-incorporated model assumptions (section 3.2 and 3.3), and one new model assumption that we propose in this study to represent priming effect in models (i.e. section 3.4 – representing priming effect in models). We have explicitly stated the logic of the Method and Result section in Page 7, Lines 14 – 18. The paragraph starting from Page 19 line 6 describes the result of assessing priming effect using the quasi-equilibrium framework. This is only part of our result, and we believe that it should not be a result section by itself.

To avoid future confusion, we will restructure the Method and Result section by 1). Move equation derivations into an Appendix, as proposed above, and only keep a minimum set of fundamental

equations to describe the quasi-equilibrium framework; 2) Change section headers with section 3.2 renamed as "Analyses of new model assumptions using the quasi-equilibrium framework" (or something similar), then add sub-sections of 3.2.1 (i.e. N uptake representation), 3.2.2 (i.e. Potential NPP), and 3.2.3 (i.e. priming effect). We will also add another paragraph following header of section 3.2 to explicitly state the purpose of each evaluation within this section. We hope that this structural arrangement will make the Method and Result section clearer to readers.

Each figure shows important information that is not immediately obvious. It needs careful text that explains to the reader what we can learn from each figure. A general 'data dump' with virtually no explanation is never a good way to proceed, but totally unacceptable in this case, as the essence of the modelling is not immediately obvious, but the reader needs to be led through the various figures to extract the key insights gained from each.

**Responses**: We thank the reviewer for raising this important point. We will revise our manuscript wherever appropriate. But we would still like to respectfully point out that both equation derivation and graphic interpretation are fundamental part of the result, and they complement each other when interpreting the results. We have detailed texts describing the baseline model figure, and each follow-on figures brought various levels of changes to the baseline figure. We have included explanatory texts throughout the derivations to suggest how each model assumption affect the results and therefore graphs. We did not "data dump" all results without any explanation.

Some of that detail is then given in the Discussion, like page 21, line 5 onwards, but only very briefly. That reference is too brief on its own and would have needed a proper description in a results section that could then be referred to. So, the Discussion might be OK if it had an appropriate Results section. But without a Results section, the reference to the various figures is still too brief to be readily and fully understood by the reader.

So, all in all, I would regard the paper as not acceptable in its current form, but that is entirely due to the lacking Results Section and insufficient description of the modeled findings. If that can be added, and the Discussion section then be modified to appropriately refer to text in the Results Section, the paper should be able to make a really strong contribution to the literature.

**Responses**: We will expand our discussion following this suggestion. However, as we argue in the responses above, we believe that the reviewer may have misinterpreted the original Method and Result section. As we stated above, we will make some structural and content edits to improve readability of our manuscript.

Minor comments: Page 3, line 7: The authors introduce the abbreviations QE for 'quasiequilibrium'. That is unnecessary in my view and just obscures the subsequent text. 'Quasi-equilibrium' is short enough and can continue to be used throughout the paper. No need to confuse the reader by an unnecessary abbreviation.

**Responses**: Will modify the text following this suggestion.

Page 7, line 7: When the authors mention 'concentration-carbon' feedback, I assume they mean 'CO2-carbon' feedback. It would be better if that could be spelled out more explicitly as 'CO2-

carbon' or something else that would leave the reader in doubt as to what concentration is referred to.

**Responses**: Great suggestion. Will revise accordingly.

Page 7, line 22: Here, it states that in assumption 3, N uptake is modeled as a saturating function of root biomass. This makes it sound as though there were no upper limit to N uptake other than that imposed by root biomass. However, the detailed model description states that N uptake is also dependent on mineralized N, which seems like a sensible assumption. Just make sure that in the initial description of this assumption, it is also made clear that mineralized N is a co-limiting factor. Currently, that is not included and gives a misleading impression of the model assumption.

**Responses**: Again, great suggestion. Will revise accordingly.

Nothing to add to the Model Description. The text after that needs some bigger overhaul as mentioned above, and I have refrained from referring to specific details as they will hopefully be changed in a bigger re-write.

**Responses**: As we stated above, we will make significant structural changes to the Method and Result section, and we hope that the revised text will be deemed satisfactory by the reviewer.

---

## Author Comment (AC3) · 14 Feb 2019

**Response to Reviewer #2**

This study tries to establish a quasi-equilibrium (QE) analytical framework, introduced by Comins & McMurtrie (1993), for evaluating model assumptions on carbon-nitrogen interaction in influencing ecosystem responses to elevated CO2. Overall, this paper is extremely valuable for understanding a variety of assumptions in influencing model outputs of carbon and nitrogen coupling.

**Responses**: We thank the reviewer for this positive feedback on our manuscript.

I particularly like your examples on page 23 to make a point that "the QE framework can highlight where additional complexity is not valuable."

**Responses**: Thank you. This is indeed one of the key points that we would like readers to pay attentions to.

Here are a few suggestions to improve your manuscript:

First, the authors may consider improve the readability of your paper so that your message can go more miles. It is quite competent of the authors to work out all those equations in section 3. But those equations will hinder delivering your message as not all the ecologists or even modelers will go over those equations when they read your paper.

**Responses**: Indeed, this is a problem that details in Section 3 may prevent a smooth read of the manuscript. As we suggested in our response to Reviewer #1, we will move some equation derivations into an Appendix (not supplementary materials) to both comply with the journal requirement of keeping all essential elements in the main body of text, as well as improve readability of our manuscript. We hope that this will sufficiently address the issue.

In addition, would it be possible to convert Table 1 to a graph so that readers can quickly get your message? To me, Table 1 is probably the most important part of your manuscript. Even though I am familiar with the subject, it still takes me a while to go over the table. Converting it to a figure may help deliver your message faster. Moreover, the abstract I don't think deliver the message well, especially the second half.

**Responses**: We will revise the abstract to make it more impactful. Table 1 is a summary of the literature which have adopted the quasi-equilibrium framework in the past. We presented some detailed introduction and discussion of this pool of literature in our Literature Review section (Section 2), and provided a graphic example in Supplementary Figure 1. Given the diverse set of model assumptions evaluated in the past pool of literature, as presented in Table 1, it is not easy to plot one figure to sufficiently synthesize all information. However, the general aim of including Table 1 and the Literature Review section was to demonstrate the usefulness of the quasi-equilibrium framework; they are not the key novel results that this manuscript adds into the literature.

Table 3 summarizes how different model assumptions affected plant production response to eCO2 at various time steps, which we believe are the "novel" results that this study brings. The graphic interpretations of the effect of each individual model assumption have been provided in Figures 4-6, and the table is a synthesis and numerical display of these results. We think that the combination of the Table and individual Figures is the most appropriate way of presenting our analyses. A stand-alone summary figure based on Table 3 appears very noisy, and makes it difficult for readers

to capture the detailed dynamics that each assessment brings. All our codes, including the quasiequilibrium framework and the plots, are publically accessible. Therefore, one can potentially explore how alternative plotting schemes compare using this code repository.

Second, the work by Comins & McMurtrie (1993) is great. But, during the same period in 1990s, Dr. Edward Rastetter has developed the Multiple Element Limitation (MEL) model of carbonnitrogen interactions. He published a few papers to illustrate similar principles on carbon-nitrogen interactions as revealed by G'DAY. In fact, Ed Rastetter also lumped all those assumptions (or processes) into three categories as in the first three items of your Table 1. MEL further shows the time scales at which each of the three categories of processes plays. In other words, MEL not only gives information about the equilibrium responses but also offers information about C/N interaction to influence transient dynamics. I think the authors at least should acknowledge Ed's work in your manuscript.

**Responses**: Thank you for this insightful comment. We will revise our text wherever appropriate to incorporate this insightful comment and valuable literature.

Third, it is fine that the G'DAY model offers an analytical framework to evaluate model assumptions on carbon and nitrogen interactions. However, the impacts (or sensitivity) of those assumptions evaluated by the framework depend on the ranges of the variables you changed. For example, your analysis shows that wood N:C flexibility is very important for modeling carbon and nitrogen interactions. What ranges of wood N:C did those studies change? Do those ranges realistically match observations? Lots of data are available to evaluate those ranges. In fact, several studies have evaluated the ranges of changes of those variables (e.g., Liang et al. 2016). Bringing observations into your study may require the authors to do additional work but will improve quality of your study. At least the authors should add discussion on observed vs. modeled ranges of changes.

**Responses**: Thank you again for this insightful comment. We will revise our manuscript accordingly. However, it is still our major purpose to demonstrate how one can analytically interpret consequence of a model assumption without running a model, rather than having readers to focus on how close one can match some theoretical model behaviors with a range of observations. Therefore, while we believe it is important to bridge observations with modeling, the inclusion of such an analysis may make this already heavily condensed paper more complicated. On the other hand, we will add a paragraph acknowledging this important issue in our discussion.

Forth, if the authors want to popularize the QE framework to be used by the broad community, they may develop a simpler scheme for others to use. The extensive list of those equations may make it very difficult for others to use.

**Responses**: Agree. We will revise our baseline model description section (i.e. section 3.1) in combination of comments made by Reviewer #1 to improve the readability of the baseline QE framework.

Reference: Junyi Liang, Xuan Qi, Lara Souza, and Yiqi Luo. 2016. Processes regulating progressive nitrogen limitation under elevated carbon dioxide: a meta-analysis. Biogeosciences, 13, 2689-2699.

**Responses**: This is a useful reference and we will add it wherever appropriate.

---

## Author Response (AR1)

**Response to Editor Kerkweg**

In my role as Executive editor of GMD, I would like to bring to your attention our Editorial version 1.1: http://www.geosci-model-dev.net/8/3487/2015/gmd-8-3487-2015.html. This highlights some

- 5 requirements of papers published in GMD, which is also available on the GMD website in the 'Manuscript Types' section: http://www.geoscientific modeldevelopment.net/submission/manuscript types.html
- In particular, please note that for your paper, the following requirement has not been met in the 10 Discussions paper:
  - "The main paper must give the model name and version number (or other unique identifier) in the title."

**Please provide explicitly the name (or its acronym) and the version number of the framework in the title of your revised manuscript.**

**Responses**: We thank the editor for the reminder of the journal requirement. We do acknowledge this important journal rule. However, we also would like to suggest that this manuscript is not a "Development and Technical Paper" or a "Model Experiment Description" paper, but rather, a "Methods for Assessment of Models" paper. The quasi-equilibrium framework is the name of the method to assess model

- 20 performance. It is not a conventional product or tool, but rather, a theoretical approach that can be used to analytically understand model performance without the need to run a model. Within its core, the quasiequilibrium framework is a way of thinking, and involves only several fundamental equations to evaluate how model performance change in response to different assumptions added into the model. Therefore, depending on the model assumptions it tests, number of equations will vary (as shown in our manuscript).
- 25 Here, we evaluated many different model assumptions using this quasi-equilibrium framework. And therefore, with all respect, we believe that assigning a version number may result in misinterpretation of our manuscript.

GMD is encouraging authors to provide a persistent access to the exact version of the source code used
for the model version presented in the paper. As explained in https://www.geoscientific-modeldevelopment.net/about/manuscript\_types.html the preferred reference to this release is through the use of a DOI which then can be cited in the paper. For projects in GitHub a DOI for a released code version can easily be created using Zenodo, see https://guides.github.com/activities/citable-code/ for details.

35 You may consider to upload the program code of the specifc version of the paper as a supplement or make the code and data of the exact model version described in the paper accessible through a DOI (digital object identifier). In case your institution does not provide the possibility to make electronic data accessible through a DOI you may consider other providers (eg. zenodo.org of CERN) to create a DOI. Please note that in the code accessibility section you can still point the reader to the GitHub 40 repository for the newest version even if you use a DOI for the relevant releases.

**Responses**: We have published our code via Zenodo, The DOI is 10.5281/zenodo.2574192. We have modified our code availability statement in the manuscript with this DOI update.

**Response to Reviewer #1**

5

10

25

Overall, this paper is excellent. It usefully adds to the body of experimental results based on the use of quasi-equilibrium models and further develops our understanding of the importance of key processes and assumptions.

Having said that, I was very happy with reading the paper up to page 19, line 5. Introduction and the associated discussion of past research was excellent, detailed and informative and structured to derived useful information and leading to new questions. The description of the model was also clear and gave all the relevant model components.

**Responses**: We thank the reviewer for this positive overall comment on our paper.

It was maybe a little long (34 equations), and there a danger that key model steps might have be swamped within a sea of less important ones. There might be a point in moving some of the model detail to Supplemental Information, and only presenting the key steps in the main text.

- **Responses**: We have reduced the number of equations in the main text from a total of 34 to a total of 22 by moving most of the baseline equations (the original Eq. 1-15) into the Appendix, as an integrated part of the manuscript. We think this modification improves the readability of the method and result section, as well as provides fundamental analytical details to comply with journal's requirement (i.e. "contain the
- 20 justification of the model structure in the main body of the paper", stated by the chief editor prior to our submission).

However, the remainder of the paper was not well presented. Page 19, line 6 should have started a 'Results Section' where the various figure and tables could have been presented and discussed in some detail. As it is, all the key findings were dumped here within half a page. I do not regard this as satisfactory.

**Responses**: We must respectfully point out that the reviewer may have misinterpreted our original Method and Result section. It was our deliberate intention to keep the Method and Result section together because the quasi-equilibrium framework is an analytical way of interpreting the likely impact of a model

- 30 assumption to model behaviors. Therefore, derivations of equations are fundamental and deeply linked to the analysis of the results. In our original Method and Result section, we firstly described the baseline quasi-equilibrium framework (section 3.1), then assessed several recently-incorporated model assumptions (section 3.2 and 3.3), and one new model assumption that we propose in this study to represent priming effect in models (i.e. section 3.4). We have explicitly stated the flow of work on original
- 35 Page 7, Lines 14 18. The paragraph starting from the original Page 19 line 6 describes the result of assessing priming effect using the quasi-equilibrium framework. This is only part of our result, and we believe that it should not be a result section by itself.

To improve readability, we have restructured the Method and Result section with the following 40 modifications. 1). Moved baseline equation derivations into an Appendix, as stated above; 2) Changed

section headers with section 3.2 renamed as "Analyses of new model assumptions using the quasiequilibrium framework"; 3) added sub-section headers to contain each individual new model assumptions tested within this study, i.e. section 3.2.1 - different N uptake representation, section 3.2.2 - potentialNPP approach, and section 3.2.3 - priming effect. Furthermore, we have revised the first two paragraphs

5 of the Method and Result section (i.e. Starting from L20, Page 7, clean version) to emphasis the structure of this section. Finally, we have revised our Abstract with clearer interpretations of the individual assumption tested here.

Each figure shows important information that is not immediately obvious. It needs careful text that

- 10 explains to the reader what we can learn from each figure. A general 'data dump' with virtually no explanation is never a good way to proceed, but totally unacceptable in this case, as the essence of the modelling is not immediately obvious, but the reader needs to be led through the various figures to extract the key insights gained from each.
- **Responses**: We thank the reviewer for raising this important point but do not agree that we added 15 "virtually no explanation". As we suggested earlier, both equation derivation and graphic interpretation complement each other when analyzing the results. We have detailed texts describing the baseline model figure (i.e. Figure 1), and each follow-on figures brought various levels of changes to the baseline figure. We have included explanatory texts throughout the Method and Result section to suggest how each model assumption, reflected via their analytical derivations, affect the figure and therefore the result
- 20 interpretations. Hopefully with our revised text, reduced equations and restructured sections, our result interpretations will be clearer for the reviewer.

Some of that detail is then given in the Discussion, like page 21, line 5 onwards, but only very briefly. That reference is too brief on its own and would have needed a proper description in a results section that could then be referred to. So, the Discussion might be OK if it had an appropriate Results section. But without a Results section, the reference to the various figures is still too brief to be readily and fully understood by the reader.

So, all in all, I would regard the paper as not acceptable in its current form, but that is entirely due to the lacking Results Section and insufficient description of the modeled findings. If that can be added, and the Discussion section then be modified to appropriately refer to text in the Results Section, the paper should be able to make a really strong contribution to the literature.

**Responses**: We have revised our discussion following this comment together with suggestions made by 35 Reviewer 2. In particular, we have added discussion on how the predicted plant response to eCO2 under progressive N limitation compares against a meta-analysis that bridges the progressive nitrogen limitation theory with data. We have also added discussion comparing the quasi-equilibrium framework with the multiple element limitation framework developed by Rastetter and Shaver (1992). We believe that these two additions are useful contextual information to enrich our discussion materials.

Minor comments: Page 3, line 7: The authors introduce the abbreviations QE for 'quasiequilibrium'. That is unnecessary in my view and just obscures the subsequent text. 'Quasi-equilibrium' is short enough and can continue to be used throughout the paper. No need to confuse the reader by an unnecessary abbreviation.

Responses: We have revised our manuscript by swapping "QE" with quasi-equilibrium.

Page 7, line 7: When the authors mention 'concentration-carbon' feedback, I assume they mean 'CO2carbon' feedback. It would be better if that could be spelled out more explicitly as 'CO2-carbon' or something else that would leave the reader in doubt as to what concentration is referred to.

- 10 something else that would leave the reader in doubt as to what concentration is referred to. Responses: Great suggestion. We have revised accordingly. The line now reads as: "The TEM (Sokolov et al., 2008) and CLM models (Thornton et al., 2009), which assumed inflexible stoichiometry, had a large climate-carbon feedback but a small CO2 concentration-carbon feedback, contrasting with the O-CN model (Zaehle et al., 2010), which assumed flexible stoichiometry and had a small climate-carbon 15 feedback are a large or concentration or backback."
- 15 feedback and a large CO2 concentration-carbon feedback.".

Page 7, line 22: Here, it states that in assumption 3, N uptake is modeled as a saturating function of root biomass. This makes it sound as though there were no upper limit to N uptake other than that imposed by root biomass. However, the detailed model description states that N uptake is also dependent

20 on mineralized N, which seems like a sensible assumption. Just make sure that in the initial description of this assumption, it is also made clear that mineralized N is a co-limiting factor. Currently, that is not included and gives a misleading impression of the model assumption. Responses: We have revised the text wherever possible.

25

5

Nothing to add to the Model Description. The text after that needs some bigger overhaul as mentioned above, and I have refrained from referring to specific details as they will hopefully be changed in a bigger re-write.

5

**Responses:** As we stated above, we have made significant structural changes to the Method and Result 30 section, and we hope that the revised text will be deemed satisfactory by the reviewer.

**Response to Reviewer #2**

This study tries to establish a quasi-equilibrium (QE) analytical framework, introduced by Comins & McMurtrie (1993), for evaluating model assumptions on carbon-nitrogen interaction in influencing

5 ecosystem responses to elevated CO2. Overall, this paper is extremely valuable for understanding a variety of assumptions in influencing model outputs of carbon and nitrogen coupling.

Responses: We thank the reviewer for this positive feedback on our manuscript.

- *I particularly like your examples on page 23 to make a point that "the QE framework can highlight where additional complexity is not valuable."*
- **Responses**: Thank you. This is indeed one of the key points that we would like readers to pay attentions to. We have also highlighted this important message in our revised Abstract.

Here are a few suggestions to improve your manuscript:

- 15 First, the authors may consider improve the readability of your paper so that your message can go more miles. It is quite competent of the authors to work out all those equations in section 3. But those equations will hinder delivering your message as not all the ecologists or even modelers will go over those equations when they read your paper.
- **Responses:** Indeed, this is a problem that details in Section 3 may prevent a smooth read of the manuscript. In our revision, we have created an Appendix to contain baseline quasi-equilibrium framework derivations to both comply with the journal requirement of keeping all essential elements in the main body of text, as well as improve readability of our manuscript.
- In addition, would it be possible to convert Table 1 to a graph so that readers can quickly get your 25 message? To me, Table 1 is probably the most important part of your manuscript. Even though I am familiar with the subject, it still takes me a while to go over the table. Converting it to a figure may help deliver your message faster. Moreover, the abstract I don't think deliver the message well, especially the second half.
- **Responses**: We have revised our abstract, in particular the second half, with explicit texts on the implications of the results obtained from this study.

Table 1 is a summary of the literature which have adopted the quasi-equilibrium framework in the past. We presented some detailed introduction and discussion of this pool of literature in our Literature Review section (Section 2), and provided a graphic example in Supplementary Figure 1. Given the diverse set of

35 model assumptions evaluated in the past pool of literature, as presented in Table 1, it is not easy to plot one figure to sufficiently synthesize all information. However, the general aim of including Table 1 and the Literature Review section was to demonstrate the usefulness of the quasi-equilibrium framework; they are not the key novel results that this manuscript adds into the literature.

Table 3 summarizes how different model assumptions affected plant production response to  $eCO_2$  at various time steps, which we believe are the "novel" results that this study brings. The graphic interpretations of the effect of each individual model assumption have been provided in Figures 4-6, and the table is a synthesis and numerical display of these results. We think that the combination of the Table

- 5 and individual Figures is the most appropriate way of presenting our analyses. A summary figure based on Table 3 could have been too noisy to make detailed interpretations on. All our codes, including the quasi-equilibrium framework and those used to generate the plots, are publically accessible (DOI 10.5281/zenodo.2574192). Therefore, one can potentially explore how alternative plotting schemes compare using this code repository.
- 10

Second, the work by Comins & McMurtrie (1993) is great. But, during the same period in 1990s, Dr. Edward Rastetter has developed the Multiple Element Limitation (MEL) model of carbon-nitrogen interactions. He published a few papers to illustrate similar principles on carbon-nitrogen interactions as revealed by G'DAY. In fact, Ed Rastetter also lumped all those assumptions (or processes) into three

- 15 categories as in the first three items of your Table 1. MEL further shows the time scales at which each of the three categories of processes plays. In other words, MEL not only gives information about the equilibrium responses but also offers information about C/N interaction to influence transient dynamics. I think the authors at least should acknowledge Ed's work in your manuscript.
- **Responses**: Thank you for this insightful comment. We have added a paragraph in the Discussion section to introduce and compare the multiple element limitation framework in the context of its similarity and differences against the quasi-equilibrium framework. Please refer to L 3 on Page 23 (clean version) and the text below for details.

"The multiple element limitation framework developed by Rastetter and Shaver (1992) analytically 25 evaluates the relationship between short-term and long-term plant responses to eCO2 and nutrient availability under different model assumptions. It was shown that there could be markedly difference in the short-term and long-term ecosystem responses to eCO2 (Rastetter et al., 1997; Rastetter and Shaver, 1992). More specifically, Rastetter et al. (1997) showed that the ecosystem NPP response to eCO2 appeared on several characteristic timescales: 1) there was an instantaneous increase in NPP, which results

- 30 in an increased vegetation C:N ratio, 2) on a timescale of a few years, the vegetation responded to eCO2 by increasing uptake effort for available N through increased allocation to fine roots, 3) on a timescale of decades, there was a net movement of N from soil organic matter to vegetation, which enables vegetation biomass to accumulate, and 4) on the timescale of centuries, ecosystem responses were dominated by increases in total ecosystem N, which enable organic matter to accumulate in both vegetation and soils.
- 35 Both the multiple element limitation framework and the quasi-equilibrium framework provides information about the equilibrium responses. These approaches also provide information about the degree to which the ecosystem replies on internally recycled N vs. exchanges of with external sources and sinks. The multiple element limitation framework also offers insight into the C-N interaction that influences transient dynamics. These analytical frameworks are both useful tools for making quantitative 40 assessments of model assumptions."

7

Third, it is fine that the G'DAY model offers an analytical framework to evaluate model assumptions on carbon and nitrogen interactions. However, the impacts (or sensitivity) of those assumptions evaluated by the framework depend on the ranges of the variables you changed. For example, your

- 5 analysis shows that wood N:C flexibility is very important for modeling carbon and nitrogen interactions. What ranges of wood N:C did those studies change? Do those ranges realistically match observations? Lots of data are available to evaluate those ranges. In fact, several studies have evaluated the ranges of changes of those variables (e.g., Liang et al. 2016). Bringing observations into your study may require the authors to do additional work but will improve quality of your study. At least the 10 authors should add discussion on observed vs. modeled ranges of changes
- 10 authors should add discussion on observed vs. modeled ranges of changes. Responses: Thank you again for this insightful comment. We have incorporated a paragraph in our discussion to reflect this recommendation. Please refer to L. 12 on Page 20 (clean version) and the paragraph below for details. As suggested in our added discussion text, we would like to highlight that it is still our major purpose to demonstrate how one can analytically interpret consequence of a model
- 15 assumption without running a model, rather than having readers to focus on how close one can match some theoretical model behaviors with a range of observations. In our Discussion section, some texts were already written illustrating this point (e.g. L. 9, Page 19, clean version: "Examples of models assuming constant (Thornton et al., 2007; Weng and Luo, 2008) and variable (Zaehle and Friend, 2010) plant tissue stoichiometry are both evident in the literature, and therefore, assuming all other model
- 20 structure and assumptions are similar, prediction differences could potentially be attributed to the tissue stoichiometric assumption incorporated into these models, as suggested in some previous simulation studies (Medlyn et al., 2016; Medlyn et al., 2015; Meyerholt and Zaehle, 2015; Zaehle et al., 2014). Together with more appropriate representation of the trade-offs governing tissue C-N coupling (Medlyn et al., 2015), further tissue biochemistry data is necessary to constrain this fundamental aspect of
- 25 ecosystem model uncertainty (Thomas et al., 2015)."). Therefore, while we believe it is important to bridge observations with modeling, the inclusion of such an analysis may make this already heavily condensed paper more complicated. A follow-up study could make a focused effort evaluating some casespecific data-model comparisons based on the quasi-equilibrium framework.
- 30 The added paragraph starting from L.12, Page 20: "Processes regulating the progressive nitrogen limitation under eCO2 were evaluated by Liang et al. (2016) based on a meta-analysis, which bridged the gap between theory and observations. It was shown that the expected diminished CO2 fertilization effect on plant growth was not apparent at the ecosystem scale due to extra N supply through increased biological N fixation and decreased leaching under eCO2. Here, our baseline assumption assumed fixed
- 35 N input into the system, and therefore plant available N is progressively depleted through increased plant N sequestration under eCO2, as depicted by the progressive N limitation hypothesis (Luo et al., 2004). A function that allows N fixation parameter to vary could provide further assessment of the tightness of the ecosystem N cycle process and its impact to plant response to eCO2. Furthermore, given the significant role wood N:C ratio plays in plant N sequestration, matching modelled range of wood tissue stoichiometry
- 40 with observations can provide addition level of evaluation of model performance. Our study provides a

generalizable evaluation based on the assumption that wood N:C ratio, when allowed to vary in a model, is proportional to leaf N:C ratio. Case-specific, more realistic evaluations can be performed based on the quasi-equilibrium framework to bridge models with observations."

5

**Forth, if the authors want to popularize the QE framework to be used by the broad community, they may develop a simpler scheme for others to use. The extensive list of those equations may make it very difficult for others to use.**

- **Responses**: Agree. We have revised our baseline model description in the main text to emphasize the key philosophy of the quasi-equilibrium framework. We have kept all related equation derivations in the
  - Appendix for interested readers to refer to. Moreover, as indicated earlier, we have made our code repository publically accessible, which can be used as the testbed for further model assumption analysis.

Reference: Junyi Liang, Xuan Qi, Lara Souza, and Yiqi Luo. 2016. Processes regulating progressive 15 nitrogen limitation under elevated carbon dioxide: a meta-analysis. Biogeosciences, 13, 2689-2699.

**Responses:** This is a useful reference, and we have incorporated it with a new paragraph discussing the performance of our model against the general patterns summarized in this literature, as introduced earlier (i.e. L. 12 on Page 20, clean version).

[revised manuscript text omitted]
 [5]                                                                                                                               |
| and ny. The menseerion with the photosynanetic constraint yields the very rong corn equinosa or court                                 |   | Deleted: br                                                                                                                                                                                      |
| 19                                                                                                                                    |   |                                                                                                                                                                                                  |

[revised manuscript text omitted]

- Martin, B., E., De Radwo Martin, G., Zaelle, S., Walker Mitholy, F., Badasha Reinko, A., Badas, R., Mishdov, M., Pak, B., Smith, B., Wang, Y. P., Yang, X., Crous Kristine, Y., Drake John, E., Gimeno Teresa, E., Macdonald
   Catriona, A., Norby Richard, J., Power Sally, A., Tjoelker Mark, G., and Ellsworth David, S.: Using models to
- guide field experiments: a priori predictions for the CO2 response of a nutrient- and water-limited native Eucalypt woodland, Global Change Biology, 22, 2834-2851, 2016.

Medlyn, B. E. and Dewar, R. C.: A model of the long-term response of carbon allocation and productivity of forests to increased CO2 concentration and nitrogen deposition, Global Change Biology, 2, 367-376, 1996.

- 25 Medlyn, B. E., Duursma, R. A., Eamus, D., Ellsworth, D. S., Prentice, I. C., Barton, C. V. M., Crous, K. Y., De Angelis, P., Freeman, M., and Wingate, L.: Reconciling the optimal and empirical approaches to modelling stomatal conductance, Global Change Biology, 17, 2134-2144, 2011.
- Medlyn, B. E., McMurtrie, R. E., Dewar, R. C., and Jeffreys, M. P.: Soil processes dominate the long-term response of forest net primary productivity to increased temperature and atmospheric CO2 concentration, Can. J. For. Res., 30, 873-888, 2000.

Medlyn, B. E., Robinson, A. P., Clement, R., and McMurtrie, R. E.: On the validation of models of forest CO2 exchange using eddy covariance data: some perils and pitfalls, Tree Physiology, 25, 839-857, 2005.

Medlyn, B. E., Zaehle, S., De Kauwe, M. G., Walker, A. P., Dietze, M. C., Hanson, P. J., Hickler, T., Jain, A. K., Luo, Y., Parton, W., Prentice, I. C., Thornton, P. E., Wang, S., Wang, Y.-P., Weng, E., Iversen, C. M., McCarthy,
H. R., Warren, J. M., Oren, R., and Norby, R. J.: Using ecosystem experiments to improve vegetation models, Nature Clim. Change, 5, 528-534, 2015.

Meyerholt, J. and Zaehle, S.: The role of stoichiometric flexibility in modelling forest ecosystem responses to nitrogen fertilization, New Phytologist, 208, 1042-1055, 2015.

- Meyerholt, J., Zaehle, S., and Smith, M. J.: Variability of projected terrestrial biosphere responses to elevated levels 40 of atmospheric CO2 due to uncertainty in biological nitrogen fixation, Biogeosciences, 13, 1491-1518, 2016.
- Norby, R. J., Warren, J. M., Iversen, C. M., Medlyn, B. E., and McMurtrie, R. E.: CO2 enhancement of forest productivity constrained by limited nitrogen availability, Proceedings of the National Academy of Sciences, 107, 19368-19373, 2010.

Oleson, K. W., Dai, Y. J., Bonan, G. B., Bosilovich, M., Dichinson, R., Dirmeyer, P., Hoffman, F., Houser, P., Levis, S., Niu, G.-Y., Thornton, P. E., Vertenstein, M., Yang, Z. L., and Zeng, X.: Technical description of the Community Land Model (CLM), National Center for Atmospheric Research, Boulder, Colorado, 2004.

Rastetter, E. B., Ågren, G. I., and Shaver, G. R.: RESPONSES OF N-LIMITED ECOSYSTEMS TO INCREASED
5 CO2: A BALANCED-NUTRITION, COUPLED-ELEMENT-CYCLES MODEL, Ecological Applications, 7, 444-460, 1997.

Rastetter, E. B. and Shaver, G. R.: A Model of Multiple-Element Limitation for Acclimating Vegetation, Ecology, 73, 1157-1174, 1992.

Reich, P. B. and Hobbie, S. E.: Decade-long soil nitrogen constraint on the CO2 fertilization of plant biomass, 10 Nature Climate Change, 3, 278, 2012.

- Rogers, A., Medlyn Belinda, E., Dukes Jeffrey, S., Bonan, G., Caemmerer, S., Dietze Michael, C., Kattge, J., Leakey Andrew, D. B., Mercado Lina, M., Niinemets, Ü., Prentice, I. C., Serbin Shawn, P., Sitch, S., Way Danielle, A., and Zaehle, S.: A roadmap for improving the representation of photosynthesis in Earth system models, New Phytologist, 213, 22-42, 2016.
- Sands, P.: Modelling Canopy Production. II. From Single-Leaf Photosynthesis Parameters to Daily Canopy Photosynthesis, Functional Plant Biology, 22, 603-614, 1995.
   Shi M. Ficher J. Photosynthesis Parameters in Ficher Photosynthesis Parameters in Ficher Photosynthesis Parameters in Photosynthesis Photosynthesis Parameters in Photosynthypers in Photosynthesis Parameters in Photosynthypers in Pho

[revised manuscript text omitted]

---

## Author Response (AR2)

I reviewed the previous version of the manuscript, and I am satisfied with the authors' explanation of their chosen structure and the changes they have made. That addresses my substantive previous issues, and I am happy to recommend acceptance of the paper.

There are just two minor issues:
1) In Figure 1a, the line illustrating the transient behaviour appears to go towards HIGHER N:C ratios before trending back to point B. I assume the line should actually go straight up without ever having increasing N:C ratios. I would be better if that could be drawn in such a way that the curve does never actually go to higher N:C. This is less apparent in Fig. 1b, but applies there as well.
2) Just a small presentation issue. In Fig. 5, the lines have some sharp discontinuities. I assume that is just a result of a course resolution in deriving the underlying data. It would be better if smoother lines could be generated, instead.

**Responses:**

We thank the reviewer for the positive comment on our manuscript. We have revised our figures with the specific comments the reviewer suggested. Specifically, the indicative line in Figure 1a is simply a graphical indication of the likely model behaviors; we have revised the line to make it go straight up, as suggested by the reviewer. Furthermore, the discontinuity issue in Figure 5 was revised with finer resolutions of N:C ratios as well. We therefore believe that we have sufficiently addressed all issues raised by the reviewer. We thank the reviewer and the editor again for the careful review of our manuscript.